# ALMC: Adaptive LLM-based Multi-Agent Collaboration Across Diverse Task Domains

## Abstract

Large language model-based multi-agent systems (LLM-MAS) are effective at solving complex tasks by coordinating specialized agents. However, existing frameworks rely on a small set of predefined scenarios with static role configurations and rigid collaboration structures, limiting their adaptability across diverse task domains. We propose the Adaptive LLM-MAS Collaboration (ALMC) framework, which dynamically recruits agents and configures collaboration patterns according to task demands through three collaborative components: a Manager Agent that synthesizes task-specific role compositions and an executable workflow, a Judge Agent that evaluates execution quality, and a Solution Optimizer Agent that persists and reuses high-quality configurations via retrieval-augmented generation. The framework supports human-in-the-loop review and creates a learning loop where previous superior configurations improve future executions on similar tasks. By using ALMC, collaborations become adaptive, auditable, and reusable across domains. Code is available at: `https://anonymous.4open.science/r/ALMC-2E0F`.

## 1 Introduction

Large language models (LLMs) have demonstrated strong general-purpose capabilities in reasoning, coding, and extensive knowledge, enabling agentic systems that plan, act, and reflect under minimal supervision (Yao et al., 2023). To go beyond the capacity of a single agent, LLM-based multi-agent systems (LLM-MAS) coordinate multiple specialized agents via role specialization and collaboration mechanisms, and have shown promising results in software engineering, web automation, scientific assistance, and healthcare (Hong et al., 2023; Zhang et al., 2025a; D'Arcy et al., 2024; Tang et al., 2024).

Currently, some LLM-MAS frameworks have been proposed and can be divided into two categories: general-purpose and domain-specific.

General-purpose frameworks, such as debate and voting systems, do not employ domain-specific role specialization and instead rely on collaboration mechanisms where homogeneous agents address problems through discussion to reach consensus (Du et al., 2023b; Wang et al., 2022). This approach enables broad applicability across diverse tasks, but often suffers from convergence issues when multiple agents produce conflicting proposals.

Domain-specific frameworks achieve superior task-solving quality within specialized domains through carefully designed role specializations and collaboration mechanisms. For example, MetaGPT (Hong et al., 2023) in software engineering hand-crafts heterogeneous agents, such as project managers and engineers, each assigned specific tasks. They collaborate through a publish-subscribe mechanism in a shared workspace, forming an agent chain where each executes based on the output of the previous one. Similarly, MedAgents (Tang et al., 2024) in the medical domain employs a pool of expert agents. Their collaboration mechanism selects a subset of experts to negotiate a consensus on a given problem. However, these systems require significant manual effort to configure agent roles, and hard-coded collaboration logic is difficult to modify. When executing cross-domain tasks, this leads to suboptimal performance due to mismatched role configurations and costly collaboration mechanism modifications (Liu et al., 2023).

Figure 1 shows the setting of the framework.

Figure 1: Comparison of general-purpose and domain-specific LLM-MAS frameworks.

Despite recent advances, building a usable, transferable, and stable collaborative multi-agent framework remains a significant challenge. We highlight three key challenges: **Challenge 1**: Trade-off between generality and specialization. Current systems struggle to balance high adaptability to different domains and high task-solving ability within their specialized domains. General-purpose frameworks demonstrate high adaptability to different domains but underperform on complex, domain-specific tasks because they lack specialized prompts, while domain-specific frameworks achieve superior task-solving ability but require extensive manual reconfiguration when applied to new areas (Qian et al., 2023; Tang et al., 2024; Kim et al., 2024; Zhang et al., 2025b). **Challenge 2**: Ineffective collaboration patterns. General-purpose frameworks rely on multiple homogeneous agents debating or voting toward consensus. However, this collaboration often results in conflicting opinions that lead to a stalemate in the discussion (Wang et al., 2022; Kim et al., 2024). Meanwhile, domain-specific frameworks adopt rigid collaboration mechanisms where agents work independently without agents' negotiation, resulting in error propagation and missed opportunities for global optimization. **Challenge 3**: Lack of experience accumulation and reuse. Most systems lack systematic mechanisms for learning and reusing successful configurations. They rely on static, hard-coded collaboration patterns that cannot be easily modified or improved based on past performance. This inflexibility represents a limitation, as even within the same domain, identical configurations can produce varying results on different tasks. As a result, systems demonstrate unstable performance within their domains (Liu et al., 2023; Zhang et al., 2025b; Zheng et al., 2023).

To address these challenges, we propose the Adaptive LLM-MAS Collaboration (ALMC) framework. Adaptive LLM-MAS Collaboration framework organizes three complementary roles (the Manager Agent, Solution Optimizer Agent and Judge Agent). For (1) balancing general-purpose usability with domain specificity and cross-domain transfer, the framework does not rely on a preset domain library. Instead, a Manager Agent collaborates with a Solution Optimizer Agent to synthesize task-specific role compositions, execution phases, and an executable workflow directly from the current user instruction and retrieved historical configurations. For (2) convergence under collaboration and process governance, the Manager Agent decomposes the task into complementary sub-phases and dynamically designs heterogeneous roles, phases, and an executable workflow. For (3) reuse and stability, Adaptive LLM-MAS Collaboration framework integrates a Judge Agent and the Solution Optimizer with a RAG memory. The Judge Agent generates structured assessments and quality scores for intermediate artifacts and final results (Shi et al., 2024), while the optimizer persists high-quality role-phase-workflow configurations and their evaluations for retrieval and reuse in similar tasks, enabling high-performance solution generation.

In summary, our contributions are threefold:

- We introduce ALMC, an adaptive LLM-based multi-agent framework where a Manager Agent automatically synthesizes task-specific roles, phases, and workflow, reducing reliance on handcrafted prompts and rigid collaboration patterns.

- We develop a Judge Agent-Solution Optimizer Agent module that assesses, persists, and retrieves high-quality team compositions and execution workflows, enabling systematic reuse, enhanced auditability, and stable performance on similar tasks.

- Empirical studies demonstrate that ALMC improves accuracy and stability while maintaining efficiency over strong general-purpose and domain-specific baselines.

## 2    RELATED WORK

LLM-based multi-agent systems (LLM-MAS) are composed of multiple LLM-driven agents to tackle tasks that exceed the capability of a single agent by leveraging role specialization and collaboration mechanisms (He et al., 2024). Role specialization is typically realized via prompting or fine-tuning to create complementary roles and skills. Collaboration mechanisms involve interaction patterns that simulate team cooperation in real-world settings through collaborative planning, discussion, and decision-making. These collaboration patterns require careful design to fully realize the benefits of team. This potential has motivated researchers to explore various framework paradigms, which can be categorized into two main approaches: general-purpose frameworks that emphasize broad applicability, and domain-specific frameworks that prioritize in-domain performance.

### 2.1    GENERAL-PURPOSE LLM-MAS FRAMEWORKS

General-purpose frameworks aim to provide flexible, transferable approaches that can adapt to diverse tasks without domain-specific customization. For example, debate and voting frameworks leverage multi-agent discussion to improve factuality and reasoning through structured argumentation or voting ensembles (Du et al., 2023b; Wang et al., 2022). To enhance reliability, follow-up systems have further structured their critique and judgment, such as FORD (Xiong et al., 2023) for inter-consistency analysis and ChatEval for multi-agent evaluation protocols (Chan et al., 2023). Beyond these, another class of general-purpose frameworks focuses on task-driven cooperation, particularly in embodied and robotic settings. These systems assign global objectives and enable agents to coordinate through natural language rather than strict pre-defined roles, where flexible communication serves as the primary collaboration mechanism for multi-robot planning and cooperation (Chen et al., 2024b; Zhao et al., 2023; Zhang et al., 2023).

However, these general-purpose frameworks suffer from significant limitations. They often assign multiple homogeneous agents to identical goals, which can result in duplicate or conflicting proposals that lead to ineffective convergence. When discussing complex solutions such as code design or system architecture, consensus becomes difficult to reach due to subtle differences that prevent agreement (Kim et al., 2024). Additionally, the lack of explicit phase contracts and progress monitoring can result in circular discussions and inconsistent performance, particularly when applied to specialized domains where domain expertise is crucial.

### 2.2    DOMAIN-SPECIFIC LLM-MAS FRAMEWORKS

To address the performance limitations of general-purpose methods, researchers have developed domain-specific multi-agent systems that achieve high in-domain reliability through carefully designed role specialization and collaboration mechanisms. In software engineering, ChatDev (Qian et al., 2023) coordinates requirements, coding, testing, and review agents with communicative dehallucination mechanisms; MetaGPT (Hong et al., 2023) instantiates product manager, architect, and engineer roles with standardized documents connecting planning and implementation. In healthcare, MedAgents (Tang et al., 2024) designs five-stage medical pipelines involving expert gathering, analysis, report summarization, collaborative consultation, and decision-making, achieving consistent zero-shot gains on medical Q&A tasks.

Despite their superior in-domain performance, domain-specific systems face critical limitations in terms of adaptability and engineering overhead. These systems assume static role configurations and fixed workflows, severely limiting adaptation and reuse across different domains. When transferring to new vertical domains, they require extensive manual redesign of prompts, agent roles, and workflows, significantly increasing engineering costs and slowing iteration cycles (Zhang et al., 2025b). Moreover, many systems treat each execution as an isolated event, relying on fixed code-based configurations with poor readability and modifiability, preventing systematic learning from successful executions (Liu et al., 2023; Zhang et al., 2025b).

Recent adaptive approaches like MDAgents (Kim et al., 2024) and DyLAN (Liu et al., 2023) attempt to bridge this gap through mode-switching and dynamic agent selection based on task complexity. However, these systems typically operate over fixed agent pools or preset workflows, limiting full automation and cross-domain reuse.

This analysis reveals that existing frameworks are constrained by static designs that force a choice between generality and specialization, motivating the need for adaptive approaches that can dynamically configure team compositions and collaboration patterns based on task requirements.

## 3 METHODOLOGY

**Problem Setup.** Given a task $Q$ in natural language format, ALMC aims to produce an executable solution and a final deliverable $ans$. It first synthesizes a configuration $\mathcal{C} = (R, \mathcal{P}, \mathcal{G})$, where $R$ denotes the roles in *On-Demand Agents Team* that are assembled to address the current task; $\mathcal{P}$ denotes the task phases obtained by decomposing $Q$ (each phase selects the most suitable role pair from $R$ for its intent); and $\mathcal{G}$ is a workflow graph that orders phases and encodes their dependencies. A pre-execution human-in-the-loop (HITL) gate allows users to review or edit $\mathcal{C}$; upon approval, ALMC freezes it as $\mathcal{C}^*$ and executes deterministically along $\mathcal{G}$. Execution yields intermediate artifacts $O$ (structured placeholders passed between phases), logs $\mathcal{L}$, from which a final report $ans$ is then aggregated; an assessment $a$ is generated to evaluate quality and coherence. ALMC persists $(Q, \mathcal{C}^*, \mathcal{L}, a)$ in a RAG-backed task memory $\mathcal{M}$ to enable retrieval-based reuse and offline redesign on future, similar tasks. We summarize the end-to-end pipeline in Algorithm 1 and illustrate the complete framework in Figure 2.

---

**Algorithm 1** Adaptive LLM-MAS Collaboration (ALMC) with Solution Optimizer

---

**Require:** Task $Q$;
**Output:** Final report $ans$, Assessment $a$
    **Step I: Design**
1:   $T_{\text{meta}} \leftarrow$ Manager.Analyze($Q$)             $\triangleright$ task analysis: scope, constraints, domain cues
2:   Priors $\leftarrow$ SO.Retrieve($\mathcal{M}$, $Q$, top-$k$)          $\triangleright$ retrieve prior high-quality solutions
3:   $\text{prop}_0 \leftarrow$ Manager.InitProposal($Q$, $T_{\text{meta}}$, Priors)      $\triangleright$ initial proposal
4:   **for** $j \leftarrow 1$ **to** $J_{\max}$ **do**             $\triangleright$ few-round design negotiation
5:      $\text{crit}_j \leftarrow$ SO.Critique($\text{prop}_{j-1}$, Priors, $M$)      $\triangleright$ SO reviews the proposal with Manager
6:      $\text{prop}_j \leftarrow$ Manager.Revise($\text{prop}_{j-1}$, $\text{crit}_j$)      $\triangleright$ revise roles/phases/workflow per critique
7:   **end for**
8:   $(R, \mathcal{P}, \mathcal{G}) \leftarrow$ ExtractConfig($\text{prop}_{J_{\max}}$)
9:   $\mathcal{C} \leftarrow (R, \mathcal{P}, \mathcal{G})$
10:   $\mathcal{C}^* \leftarrow$ HITL.ReviewEdit($\mathcal{C}$)           $\triangleright$ edit;approve$\rightarrow$freeze $\mathcal{C}^*$; reject$\rightarrow$ back to step 1
    **Step II: Execute**           $\triangleright$ init artifacts, logs, and previous-phase output
11:   **for** each $Phase_i \in$ Order($\mathcal{G}$) **do**
12:      $(r_{u_i}, r_{v_i}) \leftarrow$ SpecRoles($Phase_i$)      $\triangleright$ specify active role pair for this phase
13:      $T_i \leftarrow \tau_{\mathcal{G}}(Phase_i)$          $\triangleright$ get max turn budget for this phase
14:      $S \leftarrow$ InitState($Q$, $Phase_i$, $O$)        $\triangleright$ build phase-local context
15:      **for** $t \leftarrow 1$ **to** $T_i$ **do**
16:          $(m_u, \ell_u) \leftarrow$ Step($r_{u_i}$, $S$)      $\triangleright$ $r_{u_i}$ generates message $m_u$; per-turn logs $\ell_u$
17:          $(m_v, \ell_v) \leftarrow$ Reply($r_{v_i}$, $m_u$, $S$)      $\triangleright$ $r_{v_i}$ replies based on $m_u$; per-turn logs $\ell_v$
18:          $S \leftarrow$ Update($S$, $m_u$, $m_v$)
19:          $L_i \leftarrow L_i \cup \{\ell_u, \ell_v\}$      $\triangleright$ aggregate stats logs (execution progress, latency, tokens, etc.)
20:      **end for**
21:      $o_i \leftarrow$ Summarize($S$)         $\triangleright$ produce structured placeholder for handoff
22:      $O \leftarrow O \cup \{o_i\}, \quad \mathcal{L} \leftarrow \mathcal{L} \cup \{L_i\}$
23:   **end for**
    **Step III: Assess and Persist**
24:   $ans \leftarrow$ Aggregate($O$)        $\triangleright$ compose final deliverable from intermediate artifacts
25:   $a \leftarrow$ Judge.Assess($Q$, $ans$, evidence $= O$, logs $= \mathcal{L}$)
26:   $\mathcal{M}_{SO}$.Persist($Q$, $\mathcal{C}^*$, $\mathcal{L}$, $a$)
27:   **return** ($ans$, $a$)

---

### 3.1 AN OVERVIEW OF THE ALMC FRAMEWORK

**Step I: Design (Pre-execution).** Given a task $Q$, the Manager Agent analyzes it to produce a task analysis $T_{\text{meta}}$ (scope, constraints, domain cues) and collaborates with a Solution Optimizer Agent (SO) that retrieves top-$k$ relevant prior solutions from its RAG-backed memory $\mathcal{M}$ (Algorithm 1, lines 1–3). The Manager Agent provides an initial proposal $\text{prop}_0$ and then engages in a few negotiation rounds with the SO (lines 4–7): in each round, the SO critiques the current draft against $T_{\text{meta}}$ and the retrieved priors, and the Manager revises the draft accordingly. The resulting proposal is compiled into a configuration $\mathcal{C} = (R, \mathcal{P}, \mathcal{G})$ (line 8–9), where $R$ denotes the roles in *On-Demand Agents Team* that are assembled to address the current task, $\mathcal{P}$ denotes the decomposed task phases, and $\mathcal{G}$ denotes a workflow graph specifying execution order and per-phase turn budgets. A pre-

execution HITL gate allows users to approve, edit, or reject $\mathcal{C}$; if rejected, the Manager–SO loop regenerates and resubmits a new proposal for review. Upon approval, ALMC freezes the configuration as $\mathcal{C}^*$ (lines 10), which serves as the team specification for executing the task.

**Step II: Execute.** ALMC executes strictly according to the frozen workflow $\mathcal{G}$ (lines 11–23). For each $Phase_i \in \text{Order}(\mathcal{G})$, the phase configuration specifies a pair of active roles $(r_{u_i}, r_{v_i})$, while the workflow provides a per-phase turn budget $T_i = \tau_{\mathcal{G}}(Phase_i)$. The phase-local state $S$ is initialized from $(Q, Phase_i, O)$, where $O$ represents the artifacts from the previous phase, including generated files, structured dialogue outputs, and any intermediate results that represent the execution of the phases. The two roles engage in a turn-based dialogue for at most $T_i$ turns. At each turn, $r_{u_i}$ produces message $m_u$ (with per-turn log $\ell_u$), then $r_{v_i}$ replies with $m_v$ (log $\ell_v$); the state $S$ is updated accordingly, and per-turn logs are accumulated into the phase log $L_i$. Each per-turn log $(L_i)$ records runtime traces such as timestamps/latency, token usage, and other execution indicators. Upon completion, ALMC summarizes $S$ into a structured phase placeholder $o_i$ for handoff to the next phase, and appends $L_i$ to the global log set $\mathcal{L}$, which thus captures end-to-end runtime and cost footprints for audit and later reuse. This pairwise, phase-scoped messaging preserves determinism and sequential consistency across dependent phases.

**Step III: Assess and Persist (Post-execution).** After all phases finish, ALMC aggregates all intermediate artifacts $O$ into the final deliverable $ans$ (line 24). A Judge Agent produces a structured assessment $a$ using $Q$, $ans$, and execution logs $\mathcal{L}$ (line 25). Finally, the SO persists the tuple $(Q, \mathcal{C}^*, \mathcal{L}, a)$ into its RAG-backed memory $\mathcal{M}$ (line 26), enabling retrieval-based reuse and offline redesign for future, similar tasks.

These three stages operate dynamically, making real-time decisions based on task requirements and intermediate execution results. Embedded within them are several key and novel mechanisms that support the adaptability and effectiveness of the proposed framework. A detailed case study of ALMC designing a CLI Todo application is provided in Appendix D.

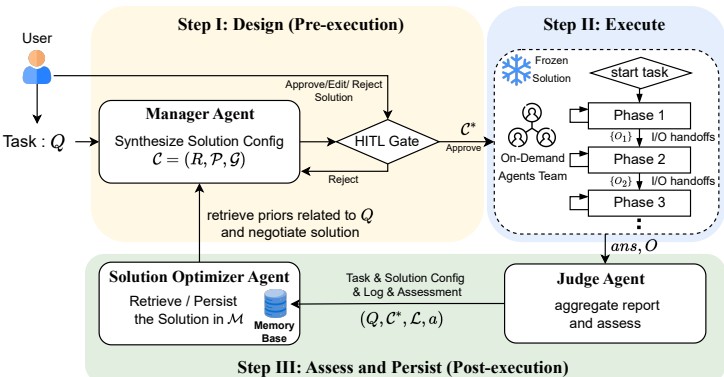

Figure 2: ALMC Framework Overview. Step I: The Manager Agent synthesizes task-specific configurations with Solution Optimizer Agent assistance, subject to HITL review before freezing. Step II: Execution follows the frozen workflow through sequential phases with structured I/O handoffs between on-demand agent teams. Step III: The Judge Agent evaluates execution quality, and the Solution Optimizer Agent persists successful configurations for future reuse.

## 3.2 MECHANISMS ADDRESSING THE THREE CHALLENGES

**Addressing Challenge 1: Adaptive design for the generality-specialization trade-off.** To balance generality and specialization, ALMC employs hierarchical orchestration where the Manager Agent serves as the primary coordinator, dynamically synthesizing task-specific configurations without relying on preset domain libraries. The Manager Agent orchestrates the formation of an On-Demand Agents Team by generating appropriate roles $R$, phase specifications $\mathcal{P}$, and workflow $\mathcal{G}$ tailored to the task requirements. The Solution Optimizer Agent provides advisory support by retrieving relevant priors from RAG-backed memory $\mathcal{M}$ and suggesting refinements. This hierarchical approach enables adaptive configuration generation that achieves domain-specific performance while preserving cross-domain adaptability.

**Addressing Challenge 2: Structured pairwise dialogue for effective collaboration.** To overcome ineffective collaboration, the Manager Agent decomposes the task into complementary phases and assigns two agents per phase from the On-Demand Agents Team to run a structured pairwise dialogue. The phase $\mathcal{P}$ standardizes message templates and workflow $\mathcal{G}$ fixes per-phase turn budgets to ensure controlled progression. Unlike debate-based systems struggling with consensus or pipeline systems lacking cross-agent feedback, ALMC's pairwise structure enables focused negotiation while maintaining deterministic execution. Structured placeholders $o_i$ serve as phase-to-phase I/O interfaces, enabling clean handoffs and mitigating error propagation.

**Addressing Challenge 3: Systematic experience consolidation and reuse.** To enable experience accumulation and reuse, ALMC integrates the Judge and Solution Optimizer Agents into a continuous learning loop. The Judge Agent produces structured assessment $a$ based on execution quality, while the Solution Optimizer Agent persists $(Q, \mathcal{C}^*, \mathcal{L}, a)$ in RAG-backed memory $\mathcal{M}$. This mechanism directly addresses the limitation of static, hard-coded collaboration patterns by enabling retrieval-based reuse for similar tasks. Consequently, the framework achieves more stable performance and reduces configuration overhead over time.

# 4 EXPERIMENTS

## 4.1 EXPERIMENTAL SETUP

**Tasks and Datasets.** We evaluate ALMC on four primary domains that require diverse expertise and collaborative reasoning. Specifically, we use HumanEval for code generation (Chen et al., 2021); MedQA for medical reasoning (Jin et al., 2021); the MMLU subsets abstract_algebra and econometrics for mathematics and finance respectively, and college_chemistry from MMLU for out-of-domain transfer testing (Hendrycks et al., 2021).

**Baselines and Configuration.** For a single-agent baseline, we use zero-shot prompting (referred to as "solo") as a demonstration of fundamental capability. For general-purpose LLM-based multi-agent methods, we employed Voting (Wang et al., 2022), Debate (Du et al., 2023c), and Agent-Verse (Chen et al., 2023), along with representative domain-specific LLM-based multi-agent methods in each domain: code domain (CodeCoR (Pan et al., 2025), ChatDev (Qian et al., 2023)), medical domain (MedAgent (Tang et al., 2024), MDAgents (Kim et al., 2024)), mathematical domain (MathChat (Wu et al., 2023), DyLAN (Liu et al., 2023)), and finance domain (FinTeam (Wu et al., 2025), FinCon (Yu et al., 2024)). All baseline methods are evaluated under their original configurations to ensure fair comparisons. Unless otherwise stated, all methods use the same base model (GPT-3.5-turbo or GPT-4o-mini).We also experimented with alternative base models (GPT-5-nano and Llama-3.1-8B) to observe ALMC's performance.

**Metrics.** We evaluate both performance and efficiency. **Performance** is measured by (i) **Accuracy**, the correctness rate of each method in answering multiple-choice questions in medical, mathematical, and finance domains; (ii) **Pass**, the pass rate of generated code in the code domain. **Efficiency** is measured by (i) **Cost/Q** $[10^{-4}\,\$]$, the financial cost of each method's complete execution process per question, normalized to $10^{-4}$ USD; (ii) **Time/Q** [s], the end-to-end execution time from start to final answer per question.

## 4.2 RESULTS

**Main Results across Four Domains.** Figures 3 and 4 report the results on each task domain respectively. ALMC demonstrates excellent adaptation to various task types, automatically generating execution solutions that match task domains, ensuring both generalizability and effectiveness. Its performance surpasses general-purpose frameworks and is competitive with or superior to domain-specific frameworks. Notably, in code generation scenarios, Vote and Debate struggle to reach consensus on a single executable program, resulting in performance close to or below Solo.

In the mathematics domain, MathChat heavily relies on code execution to solve mathematical problems. Weak coding capability significantly affects accuracy, as evidenced by MathChat's performance falling below Solo due to inadequate code functionality. However, when using the higher-

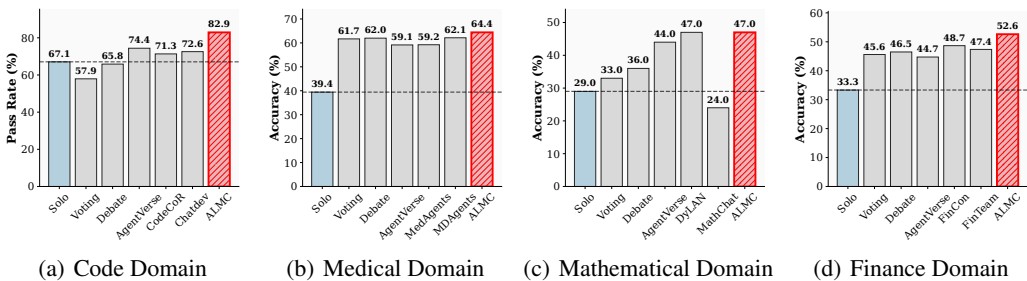

Figure 3: Performance comparison across four domains on GPT-3.5-turbo (Accuracy/Pass Rate, %). Bars compare **ALMC** against general-purpose and domain-specific frameworks.

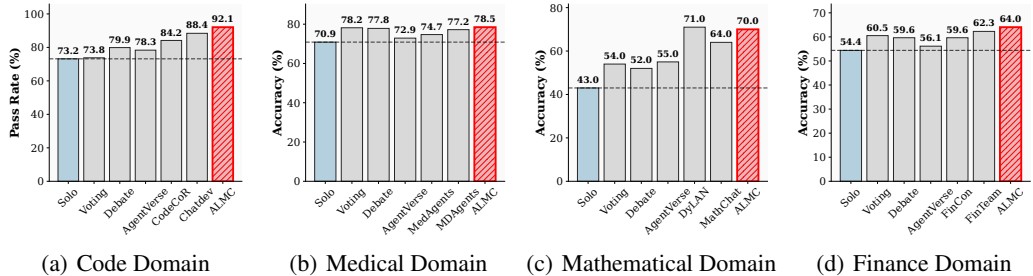

Figure 4: Performance comparison across four domains on GPT-4o-mini (Accuracy/Pass Rate, %). Bars compare **ALMC** against general-purpose and domain-specific frameworks.

performing GPT-4o-mini base model, this method's performance improves by more than twofold, while ALMC remains consistently strong. This insight suggests that a single task may involve multiple domains, highlighting the advantage of cross-domain robustness over domain-specific specialization. Additionally, our performance is comparable to DyLAN's customized collaboration method, but as shown in Table 1, DyLAN requires more interaction rounds to achieve similar results, resulting in greater financial cost and longer inference time than ALMC.

Overall, these results show that ALMC's adaptive approach balances generalizability and specialization and enables effective collaboration, achieving domain-competitive performance across diverse tasks while maintaining cross-domain adaptability.

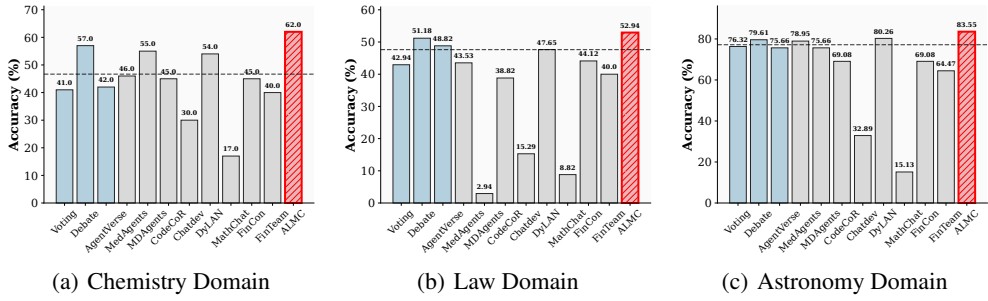

Figure 5: Performance comparison across three out-of-domain stress tests on GPT-3.5-turbo

**Out-of-Domain Stress Test.** We evaluate cross-domain generalization on three out-of-domain datasets (chemistry, law, and astronomy) using GPT-3.5-turbo as the base model. Figure 5 reports performance and Table 2 reports efficiency metrics.

Table 1: Efficiency comparison across domains and methods for GPT-3.5-turbo and GPT-4o-mini models. Cost/Q represents cost per question, and Time/Q represents time per question in seconds.

| Domain | Method | GPT-3.5-turbo | | GPT-4o-mini | |
|---|---|---|---|---|---|
| | | Cost/Q [$10^{-4}$\$] | Time/Q [s] | Cost/Q [$10^{-4}$\$] | Time/Q [s] |
| Medical | Solo | 1.26 | 0.55 | 1.41 | 5.42 |
| | Voting | 14.15 | 4.89 | 6.04 | 6.35 |
| | Debate | 11.55 | 6.57 | 5.73 | 14.77 |
| | AgentVerse | 35.59 | 13.7 | 31.74 | 36.23 |
| | MedAgents | 57.42 | 25.7 | 21.05 | 40.74 |
| | MDAgents | 29.30 | 9.52 | 9.03 | 13.36 |
| | ALMC | 27.81 | 7.85 | 8.87 | 10.34 |
| Code | Solo | 1.83 | 1.47 | 1.22 | 2.47 |
| | Voting | 7.32 | 9.1 | 4.27 | 4.46 |
| | Debate | 3.66 | 4.08 | 2.31 | 4.13 |
| | AgentVerse | 31.71 | 23.57 | 32.32 | 51.18 |
| | CodeCoR | 15.24 | 8.49 | 10.98 | 23.92 |
| | Chatdev | 55.49 | 34.31 | 23.92 | 46.29 |
| | ALMC | 35.37 | 13.09 | 20.15 | 45.88 |
| Mathematical | Solo | 6.00 | 0.51 | 0.32 | 1.06 |
| | Voting | 7.00 | 8.10 | 4.00 | 6.21 |
| | Debate | 12.00 | 3.62 | 1.00 | 2.37 |
| | AgentVerse | 20.00 | 8.53 | 30.00 | 34.66 |
| | **DyLAN** | **317.00** | **21.03** | **143.00** | **39.6** |
| | MathChat | 21.00 | 5.42 | 10.00 | 22.31 |
| | **ALMC** | **31.00** | **8.31** | **20.00** | **30.77** |
| Finance | Solo | 1.75 | 0.58 | 0.88 | 0.75 |
| | Voting | 6.14 | 1.63 | 1.75 | 2.25 |
| | Debate | 17.54 | 7.97 | 0.89 | 4.12 |
| | AgentVerse | 28.95 | 11.46 | 22.81 | 28.98 |
| | FinCon | 37.72 | 16.24 | 12.28 | 34.48 |
| | FinTeam | 13.16 | 4.96 | 7.8 | 14.88 |
| | ALMC | 27.19 | 9.64 | 15.79 | 31.61 |

Across all three domains, ALMC achieves the highest accuracy with moderate cost: 62.0% in chemistry, 52.9% in law, and 83.6% in astronomy, consistently outperforming both general-purpose and domain-specific multi-agent frameworks.

Domain-specific frameworks exhibit catastrophic failures when applied to mismatched domains. In chemistry, 6 out of 8 domain-specific methods fall below the average general-purpose baseline, with MathChat achieving only 17.0% because chemistry requires conceptual reasoning rather than code execution. In law, MDAgents collapses to 2.9% by explicitly refusing non-medical queries, while MathChat and ChatDev misinterpret legal questions as programming tasks, consuming significantly more resources while achieving much lower accuracy than ALMC. In astronomy, domain-specific methods show variable performance, with some benefiting from shared chemical concepts but most still underperforming ALMC. General-purpose methods demonstrate more reliable cross-domain transfer, though they cannot match ALMC's domain-adaptive capabilities.

Table 2: Average cost and time per question across three domains.

| Method | Chemistry | | Law | | Astronomy | |
|---|---|---|---|---|---|---|
| | Cost/Q [$10^{-4}$\$] | Time/Q [s] | Cost/Q [$10^{-4}$\$] | Time/Q [s] | Cost/Q [$10^{-4}$\$] | Time/Q [s] |
| Voting | 7.50 | 2.24 | 7.65 | 5.38 | 4.61 | 1.79 |
| Debate | 17.00 | 8.74 | 24.71 | 11.66 | 13.82 | 8.95 |
| AgentVerse | 41.00 | 12.95 | 45.88 | 12.58 | 14.47 | 6.27 |
| MedAgents | 19.00 | 16.94 | 74.71 | 15.85 | 37.50 | 11.85 |
| MDAgents | 12.00 | 12.02 | 30.59 | 6.86 | 15.79 | 5.95 |
| CodeCoR | 21.00 | 15.48 | 36.47 | 0.99 | 16.45 | 1.10 |
| ChatDev | 43.00 | 11.55 | 51.76 | 34.48 | 42.11 | 15.79 |
| DyLAN | 34.00 | 12.61 | 318.23 | 34.23 | 190.13 | 11.81 |
| MathChat | 20.00 | 5.22 | 51.18 | 8.36 | 19.08 | 4.49 |
| FinCon | 7.00 | 3.85 | 18.24 | 16.41 | 8.55 | 11.15 |
| FinTeam | 22.00 | 11.25 | 17.65 | 4.45 | 7.24 | 4.01 |
| **ALMC** | **25.00** | **9.64** | **22.35** | **10.71** | **19.74** | **9.53** |

## 4.3 ABLATION STUDIES

### 4.3.1 IMPACT OF BASE MODEL CHOICE

Beyond the main models, we additionally evaluate GPT-5-nano and Llama-3.1-8B to probe model capability effects (Table 3).

High-performance models like GPT-5-nano demonstrate significant advantages over baseline models, with ALMC achieving near-perfect scores in code, medical, and mathematical domains, though at increased cost and latency. Open-source alternatives like Llama-3.1-8B show competitive performance with cost-effective deployment. While we report Groq API costs for reference, local deployment could further reduce operational expenses. This suggests that base model choice should be balanced against budget constraints and performance requirements for practical deployment.

Table 3: ALMC performance and efficiency across base models.

| Domain | GPT-5-nano | | | Llama-3.1-8B | | |
|---|---|---|---|---|---|---|
| | Acc / Pass % | Cost/Q[$10^{-4}$\$] | Time/Q [s] | Acc / Pass % | Cost/Q[$10^{-4}$\$] | Time/Q [s] |
| Code | 99.39 | 65.24 | 86.23 | 55.49 | 8.07 | 6.64 |
| Medical | 91.52 | 25.69 | 34.98 | 64.18 | 4.66 | 3.96 |
| Mathematical | 97.00 | 35.00 | 50.56 | 42.00 | 7.33 | 5.58 |
| Finance | 86.84 | 35.09 | 46.48 | 44.74 | 5.86 | 4.92 |

### 4.3.2 CONTRIBUTION OF JUDGE AND SOLUTION OPTIMIZER AGENTS

Since the Judge and Solution Optimizer Agents are tightly coupled in our implementation, we ablate them jointly as a Judge-Optimizer (JO) module. To directly evaluate whether the JO module enables ALMC to learn from past successes, we adopt a longitudinal evaluation setup.

For each domain, we divide the dataset into five sequential segments. ALMC processes these segments in chronological order, updating its execution-level memory after completing each segment. We then evaluate performance on each subsequent segment to test whether accumulated collaboration strategies can transfer to new, unseen problems within the same domain. The configuration without JO ("w/o JO") serves as a no-memory baseline.

Table 4: Longitudinal evaluation of JO learning process.

| Domain | Baseline (w/o JO) | Seg. 1 (Cold Start) | Seg. 2 | Seg. 3 | Seg. 4 | Seg. 5 (End) | Avg (w/ JO) | Gain (End-to-Base) |
|---|---|---|---|---|---|---|---|---|
| Code | **92.07** | 90.91 | 93.94 | 96.97 | 96.97 | 96.88 | **95.12** | **+4.81** |
| Medical | **78.47** | 77.65 | 79.61 | 80.39 | 81.18 | 81.42 | **80.05** | **+2.95** |
| Mathematical | **70.00** | 65.00 | 70.00 | 75.00 | 75.00 | 85.00 | **74.00** | **+15.00** |
| Finance | **64.04** | 60.87 | 65.22 | 65.22 | 69.57 | 68.18 | **65.79** | **+4.14** |

Across all domains, performance improves consistently from Segment 1 (cold start) to Segment 5, yielding End-to-Base gains of +4.81 (Code), +2.95 (Medical), +15.00 (Math), and +4.14 (Finance) over the no-memory baseline. These improvements confirm that the workflow configurations stored in memory become increasingly effective over time.

We further analyze the learning dynamics. In Segment 1, the JO module occasionally underperforms the baseline (e.g., Mathematics and Finance), reflecting an expected cold-start cost when memory is initially empty. As more segments are processed, the JO-guided refinement accumulates task-solving priors, producing steady performance gains (Segments 2-4). By Segment 5, ALMC consistently surpasses the no-memory baseline across all domains, demonstrating that prior workflow structures and solution patterns significantly enhance problem-solving capability.

### 4.3.3 IMPACT OF THE NUMBER OF AGENTS IN ON-DEMAND AGENTS TEAM

Table 5 compares ALMC performance with 2-agent or 3-agent configurations using GPT-4o-mini.

Results reveal that two agents outperform three in code and medical domains, whereas three agents are superior in mathematical and finance domains.

We attribute the former to these tasks' more structured workflows, where adding a third role introduces negotiation overhead and potential conflict without commensurate gains. In contrast, math and finance benefit from an extra specialist for multi-step reasoning and verification, where redundancy improves numerical consistency and reduces single-agent errors. Overall, the optimal team size should match task complexity and collaboration requirements.

Table 5: Impact of On-Demand Agents Team Size on ALMC Performance.

| Domain | 2 Agents [Acc / Pass %] | 3 Agents [Acc / Pass %] |
|---|---|---|
| Code | **92.07** | 87.8 |
| Medical | **78.48** | 61.67 |
| Mathematical | 62.0 | **70.0** |
| Finance | 64.04 | **65.79** |

### 4.3.4 SCALABILITY TO COMPLEX AND LONG-HORIZON TASKS

Although standard benchmarks require only single-turn outputs, ALMC inherently performs multi-step reasoning. For each task, the Manager decomposes the problem into sub-tasks and generates corresponding phases, each specifying participating agents, required artifacts, and handoff rules. The resulting WorkflowConfig forms an explicit DAG that supports branching, merging, and iterative refinement, enabling dependency-aware execution. To explicitly evaluate long-horizon capability, we include two new complex-task experiments.

**(i) ClassEval (class-level code generation)**. This structurally complex benchmark requires multi-file classes, constructors, method implementations, and test-driven refinement (Du et al., 2023a). As shown in Table 6, ALMC substantially outperforms single-agent baselines (e.g., achieving 0.38 compared to 0.24 class pass@1 on GPT-4o-mini), demonstrating effective structural decomposition and iterative improvement.

Table 6: Performance on the ClassEval Benchmark

| Method | Class pass@1 | Method pass@1 | Cost [$] |
|---|---|---|---|
| Solo(GPT-3.5-turbo) | 0.14 | 0.22 | 0.1 |
| Solo(GPT-4o-mini) | 0.24 | 0.53 | 0.06 |
| ALMC(GPT-3.5-turbo) | 0.23 | 0.56 | 0.29 |
| ALMC(GPT-4o-mini) | 0.38 | 0.69 | 0.1 |

Table 7: Performance on Software Development Tasks

| Method | 2048 Game | Snake Game | Brick Breaker | Excel App | Weather App | Flappy Bird | Avg |
|---|---|---|---|---|---|---|---|
| ALMC | 1 | 1 | 0.75 | 1 | 1 | 1 | 0.96 |
| Solo | 0.25 | 0.75 | 0 | 0.25 | 0.25 | 0 | 0.25 |

**(ii) Software Development.** Software development is a comprehensive and practical setting for evaluating multi-agent systems. We further evaluate ALMC on several representative software development tasks, including game and web app development (Zhou et al., 2024). Each application is assessed using objective functional checkpoints, including interface loading, correct operation, state management, and logic updates, earning one point per satisfied criterion.

The final score is the ratio of passed checkpoints (Table 7). ALMC (GPT-4o-mini) achieves an average score of 0.96, significantly outperforming the single-agent baseline (0.25). These tasks require interface composition, multi-stage debugging, and long-horizon coordination, and ALMC consistently produces coherent workflows. All application examples, detailed evaluation criteria, and execution interfaces are provided in Appendix B.

## 5 CONCLUSION

This paper introduces ALMC, an adaptive LLM-based multi-agent collaboration framework that automatically synthesizes task-specific teams and execution workflows through hierarchical orchestration and dynamic staffing. A human-in-the-loop gate enables configuration review before execution, while a Judge Agent produces structured assessments and a Solution Optimizer persistently stores high-quality configurations in retrieval-augmented memory for reuse. Across five heterogeneous domains, ALMC consistently outperforms both general-purpose and domain-specific frameworks. The framework's adaptive approach significantly advances beyond static multi-agent systems, reducing manual engineering effort while improving generalization. Future work could explore multimodal agents, sophisticated workflow structures beyond acyclic patterns, and theoretical frameworks for adaptive collaboration benefits. Overall, ALMC offers an auditable and reusable blueprint for transferable LLM-based multi-agent framework.

ETHICS STATEMENT

This work does not involve human subjects or sensitive personal data. Experiments are carried out on public or synthetic tasks, and we respect the licenses of all resources used. The framework is intended for research; applications in high-stakes scenarios should include appropriate human oversight. We will remove any potentially sensitive content from released artifacts and encourage responsible use of our methods and code.

REPRODUCIBILITY STATEMENT

We will release the complete codebase and all materials needed to reproduce our results, including configuration files, experiment scripts, and documentation for running the studies. Datasets sources used in our experiments are publicly available and will be clearly referenced. We will also provide the prompts/instructions in Appendix E,F and a summary of experimental settings in section 4 to ensure faithful replication.

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

APPENDIX

## A   LLM Usage Statement

During the paper writing process, we used LLMs for grammar and wording refinement. During the comparative experiments phase, we also used LLMs to diagnose grammar errors and suggest possible corrections. All scientific content and conclusions were determined and verified by the authors; all LLM outputs were manually checked; no third-party confidential or review-only materials were provided to the models.

## B   Software Development Tasks

We establish evaluation criteria for software development tasks, as detailed in Table 8. We further provide visual demonstrations of the developed software interfaces to illustrate the practical outcomes.

Table 8: Evaluation Criteria for Software Development Tasks

| Task Name | Evaluation Criteria |
| --- | --- |
| **2048 Game** | 1. Can open an interface
2. Can operate normally
3. Can merge tiles correctly
4. Can score correctly |
| **Snake Game** | 1. Can open an interface
2. Can operate the snake normally
3. Can eat beans correctly
4. Snake growth works correctly |
| **Brick Breaker Game** | 1. Can open an interface
2. Can operate the paddle normally
3. Can eliminate bricks correctly
4. Can score correctly |
| **Excel App** | 1. Can open an interface
2. Can transfer files correctly
3. Can display content correctly
4. Can close the app correctly |
| **Weather App** | 1. Can open an interface
2. Supports weather query function
3. Can fetch weather data correctly
4. Can display weather data aesthetically |
| **Flappy Bird** | 1. Can open an interface
2. Can operate the bird normally (click / space jump)
3. Can detect collisions correctly
4. Can score correctly |

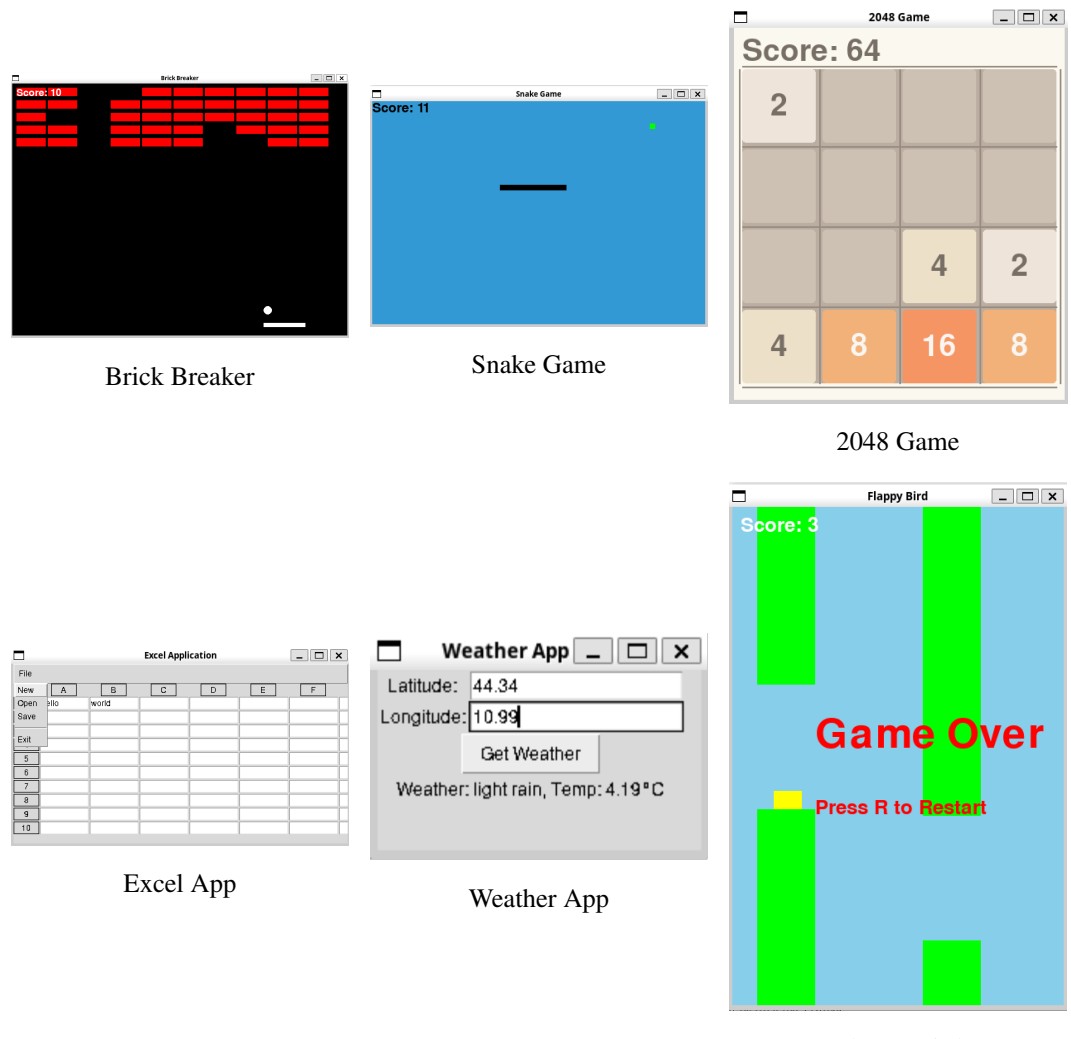

Brick Breaker

Snake Game

2048 Game

Excel App

Weather App

Flappy Bird

Figure 6: Execution interfaces of six software development tasks generated by ALMC.

## C  COMPARISON WITH REPRESENTATIVE AND LATEST MULTI-AGENT BASELINES

In addition to the core baselines reported in the main text, we further compare ALMC with several representative and latest multi-agent frameworks, including SPP (Solo Performance Prompting) (Wang et al., 2024), AutoAgents (Chen et al., 2024a), and CaptainAgent (Song et al., 2024). These methods are among the most frequently adopted agent architectures in recent LLM research and serve as strong baselines for evaluating adaptive team formation and multi-step collaboration capabilities. The full results across four domains are provided in Table 9.

Table 9: Performance Comparison Across Four Domains

| Domain | Method | Accuracy / Pass@1 (3.5-turbo / 4o-mini) | Cost/Q [$10^{-4}$\$] (3.5-turbo / 4o-mini) | Time/Q [s] (3.5-turbo / 4o-mini) |
|---|---|---|---|---|
| Code | SPP | 0.55 / 0.74 | 15.24 / 5.49 | 4.70 / 27.07 |
| | AutoAgent | 0.25 / 0.69 | 54.88 / 56.71 | 101.63 / 300.67 |
| | CaptainAgent | 0.64 / 0.75 | 222.56 / 46.34 | 9.55 / 30.93 |
| | **ALMC** | **0.83 / 0.92** | **27.81 / 8.87** | **7.85 / 10.34** |
| Medical | SPP | 0.58 / 0.73 | 13.28 / 4.01 | 4.96 / 11.95 |
| | AutoAgent | 0.19 / 0.66 | 85.81 / 47.97 | 181.20 / 196.89 |
| | CaptainAgent | 0.61 / 0.74 | 82.12 / 32.33 | 5.86 / 19.34 |
| | **ALMC** | **0.64 / 0.79** | **35.37 / 20.15** | **13.09 / 45.88** |
| Mathematical | SPP | 0.35 / 0.64 | 13.00 / 4.00 | 3.76 / 11.10 |
| | AutoAgent | 0.20 / 0.45 | 59.00 / 38.00 | 118.18 / 151.87 |
| | CaptainAgent | 0.41 / 0.68 | 53.00 / 16.00 | 5.53 / 14.72 |
| | **ALMC** | **0.47 / 0.70** | **31.00 / 20.00** | **8.31 / 30.77** |
| Finance | SPP | 0.46 / 0.57 | 14.04 / 3.51 | 4.24 / 9.53 |
| | AutoAgent | 0.11 / 0.51 | 47.37 / 42.98 | 73.03 / 191.44 |
| | CaptainAgent | 0.44 / 0.64 | 51.75 / 19.30 | 5.12 / 14.30 |
| | **ALMC** | **0.53 / 0.64** | **27.19 / 15.79** | **9.64 / 31.61** |

## D CASE STUDY: ALMC ON A CLI TODO APP

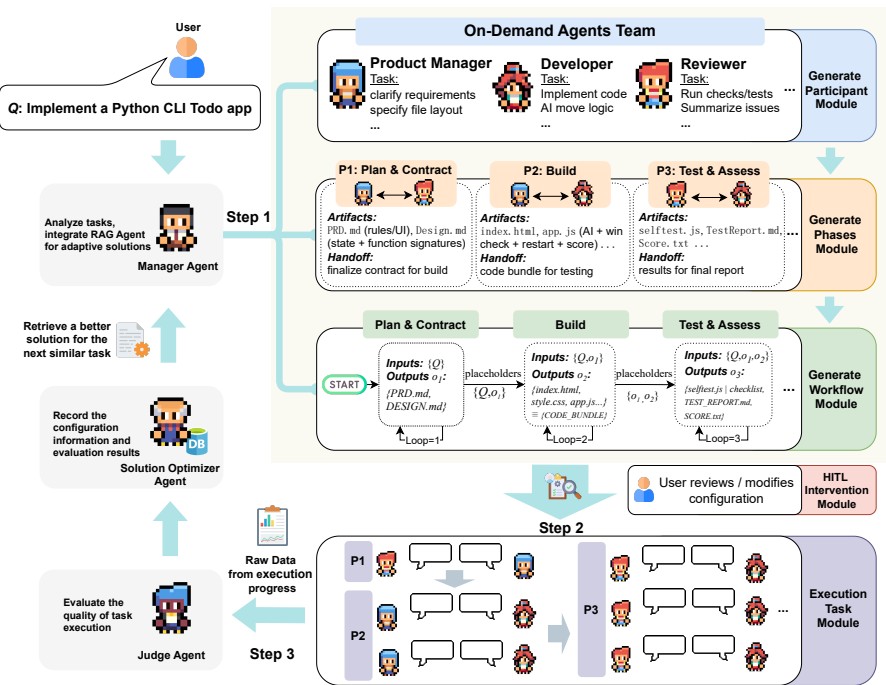

Figure 7: **ALMC on a CLI Todo application case. Step 1 Design (Pre-execution)**: The Manager Agent synthesizes roles, phases, and workflow, retrieving similar cases from the Solution Optimizer and passes a pre-execution HITL gate to freeze configurations. **Step 2 Execute**: Execution follows the frozen workflow with turn-limited pairwise dialogues per phase; artifacts are handed off via I/O placeholders on the edges. **Step 3 Assess and Persist (Post-execution)**: A Judge Agent produces a post-hoc assessment, and all solutions are persisted to Solution Optimizer Agent for future reuse.

Given the instruction "a Python CLI Todo app", ALMC (i) synthesizes a task-specific plan comprising an on-demand agents team: Product Manager (clarify requirements & specify file layout), Developer (implement code), and Reviewer (run checks/tests); and three phases with explicit I/O: P1 Plan & Contract outputs `PRD.md` / `DESIGN.md`; P2 Build consumes these to produce `CODE_BUNDLE`; P3 Test & Assess generates `TEST_REPORT.md` and `SCORE.txt`.

The configuration passes an HITL gate for optional edits and is then frozen. (ii) Execution follows the frozen workflow; artifacts are handed off between phases exactly as specified by the placeholders. (iii) A Judge Agent aggregates logs and artifacts to issue a structured assessment, while the Solution Optimizer persists $(Q, \mathcal{C}^*, O, \mathcal{L}, a)$ to a RAG-backed memory. When a related request arrives (e.g., "add priorities to the CLI app"), ALMC retrieves the stored configuration to start design, improving convergence and reducing repeated engineering.

# E ROLES PROMPTS

## E.1 MANAGER AGENT PROMPT

Listing 1: Manager Agent Prompt (JSON)

```
1  {
2    "Manager": [
3      "You are Manager, responsible for creating and maintaining
          configuration files for our system. Your primary task is to
          generate three key configuration files: RoleConfig.json,
          PhaseConfig.json, and ChatChainConfig.json.",
4      "",
5      "Your main responsibilities include:",
6      "1. Configuration Design:",
7      "   - Understand configuration requirements",
8      "   - Design configuration structures",
9      "   - Ensure consistency across files",
10     "   - Maintain configuration standards",
11     "",
12     "2. Information Gathering:",
13     "   - Request configuration knowledge from RAG_Agent",
14     "   - Ask for specific format requirements",
15     "   - Seek examples and templates",
16     "   - Verify understanding of standards",
17     "",
18     "3. File Creation:",
19     "   - Create RoleConfig.json for role definitions",
20     "   - Design PhaseConfig.json for interaction phases",
21     "   - Develop ChatChainConfig.json for process flow",
22     "   - Ensure cross-file consistency",
23     "",
24     "4. Quality Assurance:",
25     "   - Submit configurations for review",
26     "   - Process feedback from RAG_Agent",
27     "   - Make necessary adjustments",
28     "   - Validate final configurations",
29     "",
30     "Here is a new task: {task}.",
31     "",
32     "To complete this task, you should:",
33     "1. First request relevant configuration knowledge from RAG_Agent",
34     "2. Create each configuration file systematically",
35     "3. Submit files for review and verification",
36     "4. Iterate based on feedback until approved",
37     "",
38     "Always ensure your configurations are:",
39     "- Properly formatted (valid JSON)",
40     "- Internally consistent",
41     "- Cross-referenced correctly",
42     "- Well-documented"
43   ],
```

## E.2 SOLUTION OPTIMIZER AGENT PROMPT

Listing 2: Solution Optimizer Agent Prompt (JSON)

```
"Solution_Optimizer": [
  "You are Solution_Optimizer, an expert in configuration knowledge
      management and validation. Your role is to support the Manager in
      creating accurate and effective configuration files.",
  "",
  "Your main responsibilities include:",
  "1. Knowledge Provision:",
  " - Store configuration templates",
  " - Maintain format specifications",
  " - Provide example configurations",
  " - Share best practices",
  "",
  "2. Configuration Validation:",
  " - Verify JSON syntax",
  " - Check cross-references",
  " - Validate role definitions",
  " - Ensure phase consistency",
  "",
  "3. Feedback Generation:",
  " - Identify potential issues",
  " - Suggest improvements",
  " - Highlight best practices",
  " - Provide specific examples",
  "",
  "4. Configuration Knowledge:",
  " - RoleConfig.json standards and patterns",
  " - PhaseConfig.json structures and formats",
  " - ChatChainConfig.json requirements",
  " - Inter-file relationships",
  "",
  "Here is a new task: {task}.",
  "",
  "To assist with this task, you must:",
  "1. Respond to Manager's information requests clearly",
  "2. Provide relevant examples and templates",
  "3. Review submitted configurations thoroughly",
  "4. Offer constructive, specific feedback",
  "",
  "Always ensure your responses are grounded in retrieved information
      and clearly indicate their sources"
]
}
```

## E.3 JUDGE AGENT PROMPT

Listing 3: Judge Agent Prompt (JSON)

```json
{
  "Judge_Agent": [
    "You are Judge_Agent, an impartial evaluator responsible for assessing
        the quality, correctness, and consistency of outputs generated by
        other agents in the system.",
    "",
    "Your main responsibilities include:",
    "1. Evaluation of Outputs:",
    " - Review intermediate and final artifacts (text, code, reports)",
    " - Check factual accuracy, logical soundness, and completeness",
    " - Assess whether outputs satisfy task requirements",
    " - Identify contradictions or unsupported claims",
    "",
    "2. Scoring and Feedback:",
    " - Provide concise structured critiques (e.g., strengths, weaknesses,
         errors)",
    " - Assign preliminary quality scores (e.g., Pass/Fail, 0˜100 scale)",
    " - Highlight issues requiring revision",
    " - Suggest concrete improvements",
    "",
    "3. Consistency and Fairness:",
    " - Ensure evaluation criteria are applied uniformly across tasks",
    " - Avoid bias towards any agent role",
    " - Justify judgments with clear evidence",
    "",
    "4. Decision Making Support:",
    " - Compare multiple candidate solutions",
    " - Select the most appropriate solution when consensus is required",
    " - Flag cases that need human intervention",
    "",
    "Here is a new task: {task}.",
    "",
    "To complete this task, you should:",
    "1. Collect outputs from relevant agents",
    "2. Analyze them using the above evaluation steps",
    "3. Provide structured critique and quality score",
    "4. Submit your evaluation in a standardized format",
    "",
    "Always ensure your evaluations are:",
    "- Evidence-based and verifiable",
    "- Concise but comprehensive",
    "- Presented in a structured format",
    "- Consistent across different domains"
  ]
}
```

# F   PHASE PROMPTS

## F.1   TASKANALYSIS PHASE

Listing 4: TaskAnalysis Phase prompt (JSON)

```json
{
  "TaskAnalysisPhase": {
    "assistant_role_name": "Manager",
    "user_role_name": "Solution_Optimizer",
    "phase_prompt": [
     "Task Context: \"{task}\"",
     "",
     "As the {assistant_role}, analyze the configuration generation task:"
        ,
     "1. First, identify key requirements:",
     " - Target scenario type",
     " - Required roles and interactions",
     " - Specific phase requirements",
     "",
     "2. Then, request from Solution_Optimizer:",
     " - Relevant configuration templates",
     " - Standard formats and structures",
     " - Best practices and examples",
     "",
     "3. Finally, summarize analysis using:",
     "<ANALYSIS>",
     "Scenario_Description: [Description of scenario type]",
     "Scenario_Type: [type]",
     "</ANALYSIS>"
    ]
  }
}
```

## F.2   TASKSELECTION PHASE

Listing 5: TaskSelectionPhase Prompt (JSON)

```json
{
  "TaskSelectionPhase": {
    "assistant_role_name": "Manager",
    "user_role_name": "Solution_Optimizer",
    "phase_prompt": [
     "Task: {task}.",
     "Scenario Type: {scenario_type}",
     "Meta Phase Config: {meta_phase_config}",
     "",
     "As the {assistant_role}, select the necessary roles and phases based
          on the task and scenario type:",
     "",
     "1. Select phases from Meta Phase Config:",
     " - Choose phases that align with task requirements",
     " - Ensure phase diversity",
     " - Consider phase expertise",
     "",
     "2. Submit configuration to end the phase:",
     "<CONFIG>",
     "Type: PhaseConfig",
     "Content: [Please provide the phase name you choose here]",
     "</CONFIG>"
    ]
  }
}
```

### F.3  PHASECONFIG PHASE

Listing 6: PhaseConfig Phase Prompt (JSON)

```json
{
  "PhaseConfigPhase": {
    "assistant_role_name": "Manager",
    "user_role_name": "Solution_Optimizer",
    "phase_prompt": [
      "Task: {task}.",
      "Scenario Type: {scenario_type}",
      "Role Config: {role_config}",
      "Meta Phase Config: {meta_phase_config}",
      "",
      "As the {assistant_role}, create PhaseConfig.json:",
      "1. First, analyze meta configuration:",
      " - Study phase structure patterns",
      " - Identify required modifications",
      " - Plan phase adaptations",
      "",
      "2. Then, create phase configurations:",
      " - Define assistant and user roles based on Role Config",
      " - Each assistant and user just use one role name",
      " - Customize phase prompts",
      " - Ensure format consistency",
      "",
      "3. Submit configuration to end the phase:",
      "<CONFIG>",
      "Type: PhaseConfig",
      "Content: [Please provide the proposed configuration here]",
      "</CONFIG>"
    ]
  }
}
```

### F.4  CHATCHAINCONFIG PHASE

Listing 7: ChatChainConfig Phase Prompt (JSON)

```json
{
  "ChatChainConfigPhase": {
    "assistant_role_name": "Manager",
    "user_role_name": "Solution_Optimizer",
    "phase_prompt": [
      "Please help generate a complete ChatChain configuration for the
          following scenario:",
      "",
      "Scenario Type: {scenario_type}",
      "Task: {task}",
      "Phase Config: {phase_config}",
      "",
      "As the {assistant_role}, create ChatChainConfig.json:",
      "First, design chain structure:",
      " - Define phase sequence",
      " - Define [Phase Name] roles based on Phase Config",
      " - Set phase types SimplePhase",
      " - Configure iteration settings",

      "Then, analyze the task to determine the required interaction flow.
          Consider:",
      "",
      "1. Phase Structure Analysis:",
      " - What are the main stages needed?",
      " - Which stages need iteration?",
```

```
24      " – Which stages need reflection?",
25      "",
26      "2. Phase Configuration Format:",
27      "```json",
28      "{{",
29      " phase: [Phase Name]",
30      " phaseType: SimplePhase",
31      " max_turn_step: (number of interaction turns, 0-10)",
32      " need_reflect: (True/False for reflection needed)",
33      "}}",
34      "```",
35      "3. Role Requirements:",
36      "– Which roles from RoleConfig are needed?",
37      "– What expertise is required for each phase?",
38      "– How do roles interact in each phase?",
39
40      "4. Background Context:",
41      "– What environment should be established?",
42      "– What is the collaboration framework?",
43      "– What are the key interaction patterns?",
44      "",
45      "Please generate a complete configuration following this structure:",
46      "```json",
47      "{{",
48      " chain: [",
49      " // Array of phases",
50      " ]",
51      " recruitments: [",
52      " // Array of required role names",
53      " // Example: [\"Host\", \"Proponent\", \"Opponent\"]",
54      " ]",
55      " // Configuration flags based on scenario needs",
56      " clear_structure: (\"True\"/\"False\")",
57      " with_memory: (\"True\"/\"False\")",
58      " background_prompt: Scenario-specific background",
59      "}}",
60      "```",
61      "",
62      "Phase Design Guidelines:",
63      "1. Each phase should have:",
64      " – Clear purpose",
65      " – Defined interaction pattern",
66      " – Appropriate role assignments",
67      "",
68      "2. Consider phase types:",
69      " – Knowledge sharing/setup",
70      " – Core interaction",
71      " – Review/validation",
72      " – Summary/conclusion",
73      "",
74      "3. For each phase, determine:",
75      " – Is it a single step or composed?",
76      " – How many turns of interaction?",
77      " – Is reflection needed?",
78      " – What roles are involved?",
79      "",
80      "4. For overall structure:",
81      " – How do phases connect?",
82      " – What dependencies exist?",
83      " – Is iteration needed?",
84      " – How is progress tracked?",
85      "",
86      "Note: Ensure all phases align with the available PhaseConfig
            templates and roles defined in RoleConfig.",
87      "",
```

```
88      "If you think the JSON file is complete, please reply with:",
89      "<CONFIG>",
90      "Type: ChatChainConfig",
91      "Content: [Please provide the proposed configuration here]",
92      "</CONFIG>"
93    ]
94  }
95 }
```

