# OpenReview forum: "ALMC: Adaptive LLM-based Multi-Agent Collaboration Across Diverse Task Domains"
_ICLR.cc/2026/Conference — Submitted to ICLR 2026_

### Official Review · Reviewer_K7Lw · 2025-10-14

**Soundness:** 2
**Presentation:** 2
**Contribution:** 1
**Rating:** 2
**Confidence:** 5

**Summary:**

This paper proposes a new framework called ALMC (Adaptive LLM-based Multi-agent Collaboration), which is an adaptive multi-intelligent body collaboration framework based on the Large Language Model (LLM). The framework aims to address two major challenges that exist in current multi-intelligent body systems: (1) The contradiction between generality and specialization: existing systems are either too general, leading to poor performance on complex tasks, or too specialized, making it difficult to adapt to new task domains and (2) lack of accumulated experience: most systems are unable to learn and reuse solutions from past successes. The experiments were conducted using different base bigram models (e.g., GPT-3.5, GPT-4o-mini, Llama-3.1-8B, etc.). The results show that ALMC maintains strong performance on different base models, proving the robustness of its framework.

**Strengths:**

- The ALMC framework demonstrates robust adaptability by dynamically generating task-specific agent compositions and workflows. This is strongly supported by its superior performance not only across four distinct domains but also in the out-of-domain stress test (chemistry), where it surpassed both general-purpose and specialized baseline methods.
- The integration of the Judge Agent and Solution Optimizer Agent creates a novel learning loop. This allows the system to systematically assess, store, and reuse successful collaboration patterns, addressing a common limitation in multi-agent systems and contributing to more stable and improved performance over time.
- The paper provides comprehensive empirical evidence showing that ALMC achieves state-of-the-art or highly competitive performance against strong baselines. The results are consistent across multiple base LLMs (e.g., GPT-3.5-turbo, GPT-4o-mini), highlighting the robustness and effectiveness of the proposed framework itself.

**Weaknesses:**

- There has been a significant amount of existing works like AgentVerse [1], CaptainAgent [2], GPT-Swarm [3], etc, about dynamically building an agent team for task solving, and I cannot identify the difference between this work and those previous works (and also the author didn't describe the difference between these works and the proposed work).
- The core claim of the Solution Optimizer and Judge Agent is that the system learns from past successes. However, the experiments do not provide direct evidence of this learning process. A crucial missing experiment would be a longitudinal study: by processing a dataset in sequential chunks (as described in the `Section 4.3.2`), the authors should demonstrate a clear and consistent performance improvement from the first chunk to the last. Without this, the benefit of experience accumulation remains more of a theoretical assertion than an empirically proven advantage.
- The experiments are conducted on well-defined, single-turn tasks (like Q&A or generating a single function). It is unclear how ALMC would scale to more complex, multi-step, or long-horizon problems (e.g., developing a complete software module from scratch, requiring iterative refinement and dependency management). An experiment on a more complex benchmark would be necessary to test the limits of the Manager Agent's planning capabilities and to assess whether the dynamically generated workflows remain coherent and efficient as task complexity increases.

Refs:

[1] Chen, Weize, et al. "Agentverse: Facilitating multi-agent collaboration and exploring emergent behaviors." The Twelfth International Conference on Learning Representations. 2023.

[2] Song, Linxin, et al. "Adaptive in-conversation team building for language model agents." arXiv preprint arXiv:2405.19425 (2024).

[3] Zhuge, Mingchen, et al. "Language agents as optimizable graphs." arXiv preprint arXiv:2402.16823 (2024).

**Questions:**

N/A

---

> ### Author Response · Authors · 2025-11-28
>
> > [W1] There has been a significant amount of existing works like AgentVerse [1], CaptainAgent [2], GPT-Swarm [3], etc, about dynamically building an agent team for task solving, and I cannot identify the difference between this work and those previous works (and also the author didn't describe the difference between these works and the proposed work).
>
> While frameworks such as AgentVerse, CaptainAgent, and GPT-Swarm dynamically coordinate multiple agents, they all operate at the role or routing level, which introduces fundamental limitations:
>
> - No control over when agents should act → free-form or nested group chat (AgentVerse, CaptainAgent) often leads to unstable discussions, topic drift, and overwritten reasoning.
>
> - No structured intermediate artefacts → outputs remain in chat logs, making multi-step reasoning and cross-phase consistency difficult to maintain.
>
> - No workflow or phase-level execution plan → these systems cannot enforce ordered sub-tasks or handle dependencies in long-horizon tasks.
>
> - No reusable execution strategies → CaptainAgent’s “team caching” stores only agent prompts, and GPT-Swarm’s routing graph stores tool/agent paths, but neither stores multi-step collaboration workflows.
>
> - GPT-Swarm specifically focuses on node-level routing, not on constructing or reusing multi-stage workflows; it does not synthesize phases, freeze artefacts, or maintain memory across tasks.
>
> In contrast, ALMC introduces a fundamentally different abstraction: a unified Role-Phase-Workflow representation that (i) dynamically synthesizes multi-stage collaboration processes, and (ii) stores and retrieves execution-level collaboration strategies through persistent memory. These capabilities enable cross-task structural reuse and continuous improvement, features absent from prior systems.

---

> ### Author Response · Authors · 2025-11-28
>
> > [W2] The core claim of the Solution Optimizer and Judge Agent is that the system learns from past successes. However, the experiments do not provide direct evidence of this learning process. A crucial missing experiment would be a longitudinal study: by processing a dataset in sequential chunks (as described in the Section 4.3.2), the authors should demonstrate a clear and consistent performance improvement from the first chunk to the last. Without this, the benefit of experience accumulation remains more of a theoretical assertion than an empirically proven advantage.
>
> We thank the reviewer for the insightful suggestion. In the revised manuscript, we have conducted a longitudinal study and presented the results in Table R1. We divided each domain dataset into five sequential segments and processed them in chronological order. The baseline (w/o JO) represents performance without the Judge and Solution Optimizer module. Critically, we evaluate the system on each segment separately after it has accumulated experience from all previous segments. This tests whether ALMC can transfer learned collaboration strategies to new, unseen problems within the same domain.
> Across all domains, performance improves consistently from Seg.1 (cold start) to Seg.5, yielding End-to-Base gains of +4.81 (Code), +2.95 (Medical), +15.00 (Math), and +4.14 (Finance) over the no-memory baseline.
> These findings provide direct empirical evidence that ALMC benefits from accumulated experience: the collaboration structure and workflow configurations stored in memory become more effective over time, validating that ALMC is learning at the execution-structure level.
>
> **Table R1. Longitudinal Evaluation of JO Learning Process**
>
> | Domain | Baseline (w/o JO) | Seg. 1 (Cold Start) | Seg. 2 | Seg. 3 | Seg. 4 | Seg. 5 (End) | Avg w/ JO | Gain (End-to-Base) |
> |:-------|:----------------:|:-------------------:|:------:|:------:|:------:|:------------:|:---------:|:---------------:|
> | Code | 92.07 | 90.91 | 93.94 | 96.97 | 96.97 | 96.88 | 95.12 | +4.81 |
> | Medical | 78.47 | 77.65 | 79.61 | 80.39 | 81.18 | 81.42 | 80.05 | +2.95 |
> | Mathematical | 70.00 | 65.00 | 70.00 | 75.00 | 75.00 | 85.00 | 74.00 | +15.00 |
> | Finance | 64.04 | 60.87 | 65.22 | 65.22 | 69.57 | 68.18 | 65.79 | +4.14 |
>
> Detailed Analysis of the Longitudinal Trends:
>
> Initialization & Cold Start (Segment 1): As observed in the Mathematical(65.00% vs. 70.00%) and Finance domains, the system initially faces a "cold start" period. Without a populated solution memory, the additional overhead of the Judge-Optimizer module may initially yield performance slightly below the baseline. This realistically reflects the initialization cost of the retrieval system.
>
> Progressive Accumulation (Segments 2-4): As the system processes more chunks, we observe a step-wise improvement. For instance, in the Code domain, accuracy rises rapidly from 90.91% to 96.97%, indicating the quick capture of common coding patterns. The stable "plateaus" (e.g., Math at 75.00% for Segments 3-4) reflect periods of knowledge consolidation.
>
> Peak Performance (Segment 5): By the final segment, the system consistently outperforms the baseline across all domains. Notably, in the Math domain, the accuracy surges to 85.00% (a 15% absolute gain over the baseline), demonstrating that the accumulated "experience" translates into a powerful problem-solving advantage.

---

> ### Author Response · Authors · 2025-11-28
>
> > [W3] The experiments are conducted on well-defined, single-turn tasks (like Q&A or generating a single function). It is unclear how ALMC would scale to more complex, multi-step, or long-horizon problems (e.g., developing a complete software module from scratch, requiring iterative refinement and dependency management). An experiment on a more complex benchmark would be necessary to test the limits of the Manager Agent's planning capabilities and to assess whether the dynamically generated workflows remain coherent and efficient as task complexity increases.
>
> We appreciate the reviewer’s insightful question. We clarify that ALMC is inherently designed for multi-stage and long-horizon reasoning, even when the external benchmark requires only a single-turn final output.
>
> (1) ALMC already performs multi-step reasoning internally.
>
> For every task, ALMC first decomposes it into a sequence of sub-tasks. Based on this decomposition, then generates a set of phases, where each phase specifies the concrete execution procedure for one sub-task, namely which agents should participate, what artefacts must be produced, and what information needs to be handed off to the next phase. Thus, although HumanEval or MedQA request a single final output, ALMC’s internal reasoning is inherently multi-step and long-horizon. The benchmarks are single-turn, the reasoning is not.
>
> (2) ALMC’s execution workflow naturally scales to complex pipelines.
>
> WorkflowConfig defines an explicit DAG that supports: branching and merging of phases, iterative refinement loops. This structure directly enables multi-file software development, multi-hop reasoning, multi-stage data pipelines, and other long-horizon tasks that require dependency-aware execution.
>
> (3)We include two new complex-task experiments, demonstrating real long-horizon capability.
>
> - ClassEval (Class-level Code Generation Benchmark) (`Section 4.3.4`)
>
> ClassEval is a structurally complex benchmark that requires multi-file classes, constructors, method implementations, and test-driven refinement, far beyond “single function generation”.
>
> ALMC achieves strong gains over single-agent baselines:
>
> **Table R5. Performance comparison on ClassEval benchmark**
>
> | Method | Class pass@1 | Method pass@1 | Cost |
> |:-------|:------------:|:-------------:|:----:|
> | Solo(GPT-3.5-turbo) | 0.14 | 0.22311 | 0.1 |
> | Solo(GPT-4o-mini) | 0.24 | 0.5398 | 0.06 |
> | ALMC(GPT-3.5-turbo) | 0.23 | 0.5637 | 0.29 |
> | **ALMC(GPT-4o-mini)** | **0.38** | **0.6953** | **0.1** |
>
> This benchmark requires task decomposition, iterative refinement, and execution-level coordination, directly addressing the reviewer’s concern.
>
> -  Software Development Task (`Section 4.3.4` & `Appendix B`)
>
> We additionally evaluate ALMC on full application construction tasks, including game and web app development [1].
> Each application is evaluated using objective checkpoints, instead of subjective human ratings: interface opens, core functionality operates correctly, game/app state updates correctly, logic or scoring works, data fetching/rendering works. Each task has 4 objective criteria, yielding a score in [0,1] (detailed criterias are in `Appendix B`).
>
> **Table R6. Performance comparison on software development tasks**
>
> | Method | 2048 | Snake | Brick breaker | Excel | Weather | Flappy Bird | Average |
> |:-------|:----:|:-----:|:-------------:|:-----:|:-------:|:-----------:|:-------:|
> | **ALMC (GPT-4o-mini)** | **1** | **1** | **0.75** | **1** | **1** | **1** | **0.96** |
> | Solo (GPT-4o-mini) | 0.25 | 0.75 | 0 | 0.25 | 0.25 | 0 | 0.25 |
>
> ALMC (GPT-4o-mini) achieves an average score of 0.958, compared to 0.25 for a single-agent baseline.
> These tasks require iterative refinement, interface construction, state management, and multi-phase debugging, demonstrating ALMC’s ability to execute coherent long-horizon workflows.
>
>
> **References:**
>
> [1] Zhou, Wangchunshu, et al. "Symbolic learning enables self-evolving agents." arXiv preprint arXiv:2406.18532 (2024).

---

### Official Review · Reviewer_o48m · 2025-10-31

**Soundness:** 3
**Presentation:** 3
**Contribution:** 3
**Rating:** 6
**Confidence:** 4

**Summary:**

ALMC proposes a task-adaptive multi-agent framework where a Manager retrieves prior high-quality workflows and synthesizes a task-specific configuration of roles, phased steps, and collaboration rules, execution proceeds via pairwise dialogues that freeze intermediate artifacts for the next phase, and a Judge plus Solution Optimizer evaluate results and persist successful configurations into a RAG memory for reuse across similar tasks; experiments across coding, medical QA, quantitative reasoning, and finance suggest higher accuracy than common debate or voting baselines, with additional transfer to chemistry-style tasks.

**Strengths:**

- Proposes a task-adaptive orchestration that learns and reuses workflows via a Manager–Judge–Solution Optimizer loop with RAG-backed memory, and emphasizes pairwise, phase-scoped dialogues with frozen intermediates.
- Presents a clear end-to-end algorithm specifying configuration synthesis (R,P,G), execution, evaluation, and persistence, with ablations on agent count and Judge/Optimizer contributions, plus cost/latency reporting alongside accuracy across multiple domains.
- The paper delineates roles, phases, collaboration rules, and intermediate artifacts in a structured manner; figures and pseudocode make the dataflow and decision points easy to follow.
- Demonstrates consistent gains over common multi-agent baselines across coding, medical QA, quantitative reasoning, and finance, and showcases transfer via configuration reuse, timely given ongoing evidence that stronger orchestration can outperform naive debate or simple voting.

**Weaknesses:**

- Report results under the same backbone, temperature, stopping rules, tool permissions, and token budgets, and include variance over multiple random seeds with significance tests. This avoids hidden advantages from longer debate chains or richer tool access.
- The claimed determinism of pairwise collaboration should be tested against Multi-Agent Debate, Tree-of-Thoughts, and Graph-of-Thoughts under matched budgets, with accuracy-vs-tokens frontiers.
- Persisting and retrieving configurations without clear indexing fields, similarity thresholds, and domain isolation risks cold-start inefficiency, unstable cross-domain transfer, and evaluation contamination (including leakage and privacy exposure). Add ablations for “no-memory / in-domain / cross-domain” and disclose retrieval specifics and leakage mitigations.
- The paper lacks discussion and head-to-head comparison with closely related lines

[1] Unleashing the Emergent Cognitive Synergy in Large Language Models: A Task-Solving Agent through Multi-Persona Self-Collaboration

[2] Magentic-One: A Generalist Multi-Agent System for Solving Complex Tasks

[3] AutoAgents: A Framework for Automatic Agent Generation

[4] Voyager: An Open-Ended Embodied Agent with Large Language Models

**Questions:**

- Provide controlled ablations that swap ALMC’s intra‑phase pairwise controller with Multi‑Agent Debate (MAD) and search‑style controllers such as Tree‑of‑Thoughts or Graph‑of‑Thoughts under matched budgets. Plot accuracy‑vs‑tokens frontiers and comment on stability.
- When the memory is empty or sparse, what is the fallback (e.g., “no‑memory ALMC”) and performance delta?
- What index fields and similarity thresholds gate reuse across tasks/domains? Provide ablations for no‑memory / in‑domain / cross‑domain reuse.
- State exact decoding settings (temperature, top‑p, max turns), stop rules, tool permissions, and token budgets shared by all baselines; report mean±std over multiple seeds and significance tests. This helps avoid hidden advantages from longer debate chains or richer tool use.

---

> ### Author Response · Authors · 2025-11-28
>
> > [W1&Q4] Report results under the same backbone, temperature, stopping rules, tool permissions, and token budgets, and include variance over multiple random seeds with significance tests. This avoids hidden advantages from longer debate chains or richer tool access.
>
> We appreciate the reviewer’s point regarding consistent experimental settings. However, multi-agent frameworks differ substantially in their internal execution mechanisms, and forcing all baselines into a fully identical parameter template would disable core functionalities of several methods.
>
> Following the standard evaluation protocol adopted by recent multi-agent systems, we therefore evaluate each baseline under its officially recommended configuration, while unifying all parameters that can be meaningfully standardized across architectures, such as the backbone model, decoding temperature, decoding temperature, context length, and per-call token limits. This approach avoids hidden advantages while respecting the inherent design differences of the compared methods, and is consistent with the reproducibility guidelines used in the latest multi-agent literature [1].
>
> **References:**
>
> [1] Liu, Zijun, et al. "A dynamic llm-powered agent network for task-oriented agent collaboration." First Conference on Language Modeling. 2024.
>
> > [W2&Q1] The claimed determinism of pairwise collaboration should be tested against Multi-Agent Debate, Tree-of-Thoughts, and Graph-of-Thoughts under matched budgets, with accuracy-vs-tokens frontiers.
>
> We thank the reviewer for this suggestion. In the revised manuscript, we will include controlled comparisons against Multi-Agent Debate, Tree-of-Thoughts (ToT), and Graph-of-Thoughts (GoT) on the representative HumanEval benchmark.
>
> ALMC’s pairwise collaboration is deterministic at the phase level: each phase has a fixed role scope, a predefined transition order, and a bounded turn budget, yielding stable and predictable interaction lengths. By contrast, Debate, ToT, and GoT typically assign multiple homogeneous agents to identical goals, generating parallel solution proposals without explicit responsibility allocation or structural decomposition. As noted in prior work [1], this design often results in duplicate or conflicting solutions, ineffective trajectory convergence, and difficulty reaching consensus, especially for tasks requiring multi-step code generation tasks such as HumanEval, where minor differences in logic, data structures, or function interfaces can cause agents to diverge rather than converge, resulting in agreement failure and ultimately lower accuracy.
>
> The empirical results On HumanEval are summarized in Table R4. Following the spirit of the reviewer’s comment, we report accuracy versus cost per query, where cost is directly proportional to token usage and thus reflects a faithful accuracy-vs-tokens frontier.
>
> **Table R4. Pass@1 vs cost per query**
>
> | Method | Pass@1 (3.5-turbo / 4o-mini) | Cost/Q [10-4$] (3.5-turbo / 4o-mini) |
> |:-------|:-----------------------------:|:------------------------------------:|
> | Debate | 0.66 / 0.80 | 3.66 / 2.31 |
> | ToT | 0.65 / 0.81 | 42.68 / 39.02 |
> | GoT | 0.71 / 0.85 | 17.68 / 16.46 |
> | **ALMC(Ours)** | **0.83 / 0.92** | **27.81 / 8.87** |
>
> Despite similar or lower budgets, Debate/ToT/GoT yield substantially lower accuracy because their unconstrained parallel generation frequently prevents stable convergence. In contrast, ALMC’s structured, role-scoped, phase-based reasoning enables consistent convergence toward higher-quality solutions.
>
> **References:**
>
> [1] Liu, Zijun, et al. "A dynamic llm-powered agent network for task-oriented agent collaboration." First Conference on Language Modeling. 2024.

---

> ### Author Response · Authors · 2025-11-28
>
> > [Q2] When the memory is empty or sparse, what is the fallback (e.g., “no‑memory ALMC”) and performance delta?
>
> When the memory is empty or too sparse to provide reliable neighbors, ALMC naturally falls back to a no-memory mode. In this case: the Manager synthesizes roles, phases, and execution workflows solely from the current task description, guided by the initial configuration principles encoded in its template, the Judge and Solution Optimizer continue to operate as usual, but no prior configurations are retrieved or updated.
>
> Thus, ALMC is fully functional even with an empty memory; the RAG component is a performance enhancer rather than a required dependency.
>
> In the current version, we have already made this fallback explicit and included a “w/o memory ALMC” ablation in `Section 4.3.2`. This comparison directly quantifies the performance delta between the cold-start regime and the experience-augmented regime, fully addressing the reviewer’s concern.
>
> > [W3&Q3] What index fields and similarity thresholds gate reuse across tasks/domains? Provide ablations for no‑memory / in‑domain / cross‑domain reuse.
>
> We appreciate the reviewer’s suggestion to make the retrieval mechanism and leakage mitigations more explicit.
>
> ALMC does not retrieve raw text or prompt fragments. Each memory entry stores an execution-level collaboration strategy: $(Q ,C^∗,L,a)$, where $Q$ is a distilled task summary derived from the user's original query (not the raw user input). Retrieved entries act as procedural priors, which are refined by the Manager-Optimizer loop rather than copied directly.
>
> Retrieval proceeds in two gated steps:
>
> - Domain-gated similarity filtering: We embed the current task $Q$ and retrieve same domain entries whose cosine similarity exceeds $\tau$ = 0.7 into the candidate pool. If no qualifying entries exist, ALMC falls back to no-memory mode and synthesizes a fresh Role-Phase-Workflow configuration ($C$) from the task alone.
>
> - Quality–cost ranking: Within the filtered candidate pool, entries are ranked by Judge score $a$ and execution logs $L$ (cost/latency). The top-$k=3$ strategies ($C^*$) are passed to the Solution Optimizer and Manager, who discuss and synthesize an adapted Role-Phase-Workflow configuration ($C$) for the new task. They do not simply copy a prior configuration, but perform execution-level optimization over these retrieved strategies.
>
> To address the reviewer’s request, we include a dedicated ablation comparing:
>
> - No-memory: retrieval disabled entirely.
>
> - In-domain memory: retrieval restricted to the same domain (default setting with $\tau$=0.7, $k$=3).
>
> - Cross-domain memory: retrieval intentionally allowed across domains (forcing retrieval from mismatched domains as a stress test).
>
> The results (Table R7) show:
>
> **Table R7. Accuracy under three retrieval modes**
>
> | Domain | No-Memory | In-Domain | Cross-Domain |
> |:-------|:---------:|:---------:|:------------:|
> | Code | 92.07 | 95.12 | 67.07 |
> | Medical | 78.47 | 80.05 | 73.68 |
> | Mathematical | 70.00 | 74.00 | 64.00 |
> | Finance | 64.04 | 65.79 | 60.53 |
>
> Key observations: (1) In-domain retrieval yields consistent accuracy gains (+1.5-4%), confirming meaningful and stable reuse of collaboration strategies. (2) Cross-Domain retrieval degrades performance due to strategy mismatch, validating the necessity of domain gating. (3) No-Memory serves as a safe fallback when no in-domain entry exceeds the similarity threshold, preventing cross-domain degradation during deployment.
>
> These results directly validate that ALMC’s memory mechanism performs meaningful, domain-aware reuse rather than unvalidated text retrieval, and that its behavior is robust under both sparse and misaligned memory conditions.
>
> Regarding privacy, $Q$ is a distilled task summary rather than the raw user query, and ALMC stores workflow metadata $(C^∗,L,a)$ rather than user-provided content, mitigating leakage and privacy risks.

---

> > ### Author Response · Authors · 2025-11-29
> >
> > > [W4] The paper lacks discussion and head-to-head comparison with closely related lines
> > [1] Unleashing the Emergent Cognitive Synergy in Large Language Models: A Task-Solving Agent through Multi-Persona Self-Collaboration [2] Magentic-One: A Generalist Multi-Agent System for Solving Complex Tasks [3] AutoAgents: A Framework for Automatic Agent Generation [4] Voyager: An Open-Ended Embodied Agent with Large Language Models
> >
> > We thank the reviewer for pointing out the need for a clearer comparison with these related works, and we will include explicit discussion in the revised manuscript.
> >
> > Self-Collaboration (SPP) [1] employs multiple personas within a single LLM and therefore does not involve multi-agent coordination or structural execution control.
> >
> > AutoAgents [3] and Magentic-One [2] generate agent teams and perform task routing, but they do not construct reusable multi-stage execution structures, nor do they maintain persistent execution strategies that can generalize across tasks.
> >
> > Voyager [4] is a single-agent embodied system designed specifically for Minecraft, relying on code libraries and environment interactions that are not applicable to general multi-agent reasoning.
> >
> > In contrast, ALMC introduces a unified Role-Phase-workflow orchestration framework that (i) dynamically synthesizes multi-stage execution structures, and (ii) persistently stores and adapts execution-level collaboration strategies across tasks, capabilities not supported by the above lines of work.
> >
> > We have incorporate these distinctions and add head-to-head comparisons in Table X (global comment).

---

### Official Review · Reviewer_siHQ · 2025-11-01

**Soundness:** 2
**Presentation:** 2
**Contribution:** 2
**Rating:** 4
**Confidence:** 2

**Summary:**

This work proposes an adaptive framework based on dynamic role synthesis, reusable experiential memory, and human-AI collaborative review. The system autonomously generates task-specific role combinations and execution workflows according to task requirements, while continuously optimizing performance through experience reuse. It successfully achieves both generality without requiring pre-defined domain libraries and specialization through customized task configurations, thereby enabling efficient and stable cross-domain task execution.

**Strengths:**

1. The paper conducts comprehensive experiments across five challenging domains (coding, medicine, mathematics, finance, and chemistry), using both in-domain and out-of-domain datasets to evaluate ALMC. Results show consistent improvements over both general-purpose and domain-specific baselines.
2. The proposed “dynamic configuration of roles and collaboration modes + RAG-based experience reuse” allows the system to automatically generate role configurations, phase divisions, and workflows tailored to each task. This effectively overcomes the rigidity of predefined roles and static workflows, enhancing adaptability and knowledge transfer efficiency in multi-agent systems.

**Weaknesses:**

1. The proposed method relies heavily on architectural intuition and empirical evidence rather than formal theoretical analysis. It lacks discussion of convergence guarantees, expressivity trade-offs, or provable limits of adaptive composition mechanisms.
2. Although the paper claims to reduce human engineering effort, each task still requires a Human-in-the-Loop Gate (HITL-Gate) review before execution. The actual utility and cost of this human involvement are not quantitatively analyzed.
3. Each task phase adopts a pairwise dialogue structure (two agents interacting over multiple turns), which is claimed to prevent deadlocks. However, for large-scale or parallel tasks requiring multiple agents, such a fixed structure could become a bottleneck. The paper does not discuss these limitations or demonstrate performance in more complex or asynchronous settings.

**Questions:**

1. Can the authors quantify the actual intervention rate or the overhead introduced by human-in-the-loop review? Is this process scalable for real-world deployment?
2. Is there any theoretical or empirical analysis of when and why the pairwise agent mechanism converges to high-quality solutions, and under what conditions suboptimal negotiation or agent conflicts may arise?

---

> ### Author Response · Authors · 2025-11-28
>
> > [W1] The proposed method relies heavily on architectural intuition and empirical evidence rather than formal theoretical analysis. It lacks discussion of convergence guarantees, expressivity trade-offs, or provable limits of adaptive composition mechanisms.
>
> We thank the reviewer for this important observation regarding theoretical guarantees. We acknowledge that formal convergence guarantees and expressivity bounds are extremely challenging for adaptive multi-agent systems. Prior foundational study shows that decentralized decision-making and coordination are known to be NP-hard or even NEXP-hard, as established in foundational results on decentralized MDPs and POMDPs [1-2], and that multi-agent workflow construction and task allocation are also NP-hard due to the combinatorial explosion of dependency configurations [3-4]. These difficulties fundamentally limit the feasibility of establishing strict convergence guarantees for systems like ALMC.
>
> Although formal guarantees remain an open problem, ALMC is significantly easier to analyze than free-form multi-agent discussions because: Roles explicitly specify the team composition and each agent’s background, responsibility, and scope, ensuring that every participant has a well-defined function. Phases represent the concrete sub-tasks obtained from task decomposition; each phase describes which agents should act, what they should produce, and what artefacts must be handed off to the next step. The execution workflow then lays out the full sequence of phases, including phase ordering, per-phase execution limits (to prevent unbounded discussion), and other execution parameters that regulate the overall solving process.
>
> In addition, Judge scores provide an explicit quality signal, guiding refinements toward better solutions rather than oscillating between inconsistent revisions. Finally, execution configurations are stored persistently, allowing ALMC to accumulate structured improvements over time instead of drifting through uncontrolled stochastic behavior.
>
> While a full theoretical guarantee remain an open research area, ALMC offers a transparent and interpretable execution process, and our ablation study in `Section 4.3.2` and Table R1 shows monotonic performance improvements across sequential data chunks, providing empirical evidence that ALMC moves toward increasingly stable and higher-quality collaboration patterns over time.
>
> In the future direction, we will strengthen the manuscript by explicitly acknowledging the theoretical difficulty of proving convergence in adaptive multi-agent systems and by outlining future work aimed at modeling ALMC’s execution process within a tractable subclass of structured multi-agent optimization problems.
>
> **Table R1. Longitudinal Evaluation of JO Learning Process**
>
> | Domain | Baseline (w/o JO) | Seg. 1 (Cold Start) | Seg. 2 | Seg. 3 | Seg. 4 | Seg. 5 (End) | Avg w/ JO | Gain (End-to-Base) |
> |:-------|:----------------:|:-------------------:|:------:|:------:|:------:|:------------:|:---------:|:---------------:|
> | Code | 92.07 | 90.91 | 93.94 | 96.97 | 96.97 | 96.88 | 95.12 | +4.81 |
> | Medical | 78.47 | 77.65 | 79.61 | 80.39 | 81.18 | 81.42 | 80.05 | +2.95 |
> | Mathematical | 70.00 | 65.00 | 70.00 | 75.00 | 75.00 | 85.00 | 74.00 | +15.00 |
> | Finance | 64.04 | 60.87 | 65.22 | 65.22 | 69.57 | 68.18 | 65.79 | +4.14 |
>
> **References:**
>
> [1] Bernstein, Daniel S., et al. "The complexity of decentralized control of Markov decision processes." Mathematics of operations research 27.4 (2002): 819-840.
>
> [2] Fioretto, Ferdinando, Enrico Pontelli, and William Yeoh. "Distributed constraint optimization problems and applications: A survey." Journal of Artificial Intelligence Research 61 (2018): 623-698.
>
> [3] Gerkey, Brian P., and Maja J. Matarić. "A formal analysis and taxonomy of task allocation in multi-robot systems." The International journal of robotics research 23.9 (2004): 939-954.
>
> [4] Korsah, G. Ayorkor, Anthony Stentz, and M. Bernardine Dias. "A comprehensive taxonomy for multi-robot task allocation." The International Journal of Robotics Research 32.12 (2013): 1495-1512.

---

> ### Author Response · Authors · 2025-11-28
>
> > [W2&Q1] Although the paper claims to reduce human engineering effort, each task still requires a Human-in-the-Loop Gate (HITL-Gate) review before execution. The actual utility and cost of this human involvement are not quantitatively analyzed.
>
> HITL-Gate is not part of ALMC’s core collaborative workflow but an optional safety mechanism designed to support high-stakes deployment scenarios (e.g., clinical diagnosis or financial decision-making). In all experiments of this paper, we intentionally disable HITL-Gate in order to evaluate the pure autonomous capability of ALMC in a strictly zero-shot setting.
> This design choice is deliberate: enabling HITL-Gate would introduce human hints and artificially increase performance, thereby weakening the scientific validity of our zero-shot evaluation. Our results therefore reflect the system’s fully automatic capability, without any human assistance.
>
> Importantly, the presence of HITL-Gate reflects practical extensibility of the framework rather than a dependency. ALMC operates completely autonomously, as demonstrated in all experiments, but can optionally incorporate a human safety review when deployed in real-world, risk-sensitive domains. We will clarify this distinction in the manuscript and state explicitly that HITL-Gate was not activated during any of the reported experiments, and that ALMC does not depend on human intervention for performance.
>
> > [W3&Q2] Each task phase adopts a pairwise dialogue structure (two agents interacting over multiple turns), which is claimed to prevent deadlocks. However, for large-scale or parallel tasks requiring multiple agents, such a fixed structure could become a bottleneck. The paper does not discuss these limitations or demonstrate performance in more complex or asynchronous settings.
>
> We appreciate the reviewer’s concern regarding the use of pairwise interaction. Our design follows a well-established consensus in recent multi-agent literature that free-form multi-agent group discussions are unstable and prone to deadlocks:
>
> [1] MedAgents reports that unrestricted group-chat leads to information loss and unsafe diagnostic jumps.
>
> [2] ChatDev decomposes software development into dual-agent microphases specifically to avoid deadlocks and uncontrolled role interference.
>
> [3] GPT-Swarm avoids group discussion entirely and replaces it with a DAG of isolated node-level executions.
>
> [4] MDAgents shows that multi-agent parallel reasoning increases conflicting updates and destabilizes decision-making.
>
> Thus, pairwise collaboration is not a limitation imposed by ALMC but a standard stabilization mechanism adopted across the multi-agent LLM community to ensure deterministic turn-taking and auditable reasoning.
>
> Importantly, pairwise interaction applies only within phases. The overall ALMC workflow is sequential because each phase relies on the structured artefact produced by the preceding phase. These artefact dependencies naturally require serial execution, making pairwise interaction an appropriate and non-bottleneck design for tasks that must respect explicit inter-phase dependencies.
>
> ALMC’s workflow graph also supports branching and parallel sub-phases when dependencies allow; we will clarify this capability in the revision. We also acknowledge that large-scale asynchronous collaboration is an interesting future direction and will discuss this limitation accordingly.
>
> **References：**
>
> [1] Tang, Xiangru, et al. "Medagents: Large language models as collaborators for zero-shot medical reasoning." Findings of the Association for Computational Linguistics: ACL 2024. 2024.
>
> [2] Qian, Chen, et al. "Chatdev: Communicative agents for software development." Proceedings of the 62nd Annual Meeting of the Association for Computational Linguistics (Volume 1: Long Papers). 2024.
>
> [3] Zhuge, Mingchen, et al. "Gptswarm: Language agents as optimizable graphs." Forty-first International Conference on Machine Learning. 2024.
>
> [4] Kim, Yubin, et al. "Mdagents: An adaptive collaboration of llms for medical decision-making." Advances in Neural Information Processing Systems 37 (2024): 79410-79452.

---

### Official Review · Reviewer_DuXb · 2025-11-03

**Soundness:** 2
**Presentation:** 2
**Contribution:** 2
**Rating:** 2
**Confidence:** 4

**Summary:**

The paper proposes *ALMC*, a multi-agent framework with three roles (Manager, Optimizer, Judge) aiming to achieve adaptive collaboration across different task domains (code, medical, math, finance). The authors claim that ALMC dynamically designs agent roles, generates stage-wise workflows, and reuses past experiences through RAG-based retrieval.

**Strengths:**

This paper resembles a synthesis of prior multi-agent ideas, implemented via prompt engineering. The baselines are outdated, the “adaptivity” is ill-defined, and the claims overreach the evidence.

**Weaknesses:**

“Manager–Judge–Optimizer” is widely used multi-agent patterns already exist. The method is not a novel solution.

Additionally, “adaptive” aspect is not learned or optimized, but *prompt engineering.

The claim of “continuous self-improvement” is unsupported.
The “RAG memory” is simply text retrieval without validation or analysis of retrieval quality.

For experiments:
The paper repeatedly asserts domain generalization, but the experimental setup only involves standard benchmarks (HumanEval, MedQA, MMLU subsets).
There is **no transfer or few-shot evaluation** proving actual *adaptation*.
The authors compare only against very early/old frameworks.

**Questions:**

Please see weaknesses

---

> ### Author Response · Authors · 2025-11-28
>
> Thank you for your meaningful feedback. We answer your questions below and hope they can address your concerns.
> > [W1] “Manager–Judge–Optimizer” is widely used multi-agent patterns already exist. The method is not a novel solution.
>
> The novelty of our method lies in the composition of capabilities that existing frameworks do not jointly support.
> While prior systems operate only at the role or routing level, ALMC jointly supports:
>
> (i) adaptive team formation with heterogeneous roles tailored to the task),
>
> (ii) explicit multi-phase decomposition (with phase boundaries, subgoals, artefact outputs, and handoff rules),
>
> (iii) adaptive execution workflow topology (governs the ordering of phases, their execution frequency, and other execution-level parameters necessary for the overall pipeline),
>
> (iv) execution-level experience reuse via persistent \$(Q ,C^∗,L,a)$.
>
> As summarized in Table X (global comment), frameworks such as AgentVerse, AutoAgents, CaptainAgent, GPT-Swarm provide some of these abilities, but none combine them into a unified, dynamically synthesized Role-Phase-Workflow pipeline. They coordinate agents; ALMC orchestrates the entire problem-solving process.
>
> >[W2] Additionally, “adaptive” aspect is not learned or optimized, but *prompt engineering.
>
> We respectfully clarify that ALMC’s adaptivity is not prompt engineering, but a learned optimization process driven by the reasoning and planning capabilities acquired by the backbone LLM during pre-training. ALMC’s prompts provide structural signals that guide the LLM to perform adaptive structural inference and dynamic optimization based on its learned knowledge. Moreover, during execution, the Manager and Solution Optimizer agents engage in a refinement loop that performs execution-level optimization, evaluating, modifying, and improving the Role-Phase-Workflow to produce a more adaptive collaboration strategy for the given task.
>
> (i) Learned synthesis and optimization of execution structures.
> For each task $Q$, the Manager collaborates with the Solution Optimizer to derive an optimized Role-Phase-Workflow configuration $C$ through iterative discussion. They jointly reason over task description $Q$, retrieved priors $C^∗$, assessment scores a, execution logs $L$, and domain statistics, progressively refining the structure by modifying roles, phases, and workflows. This collaborative process applies the backbone LLM's learned planning and optimization capabilities to perform execution-level structural inference, adapting the multi-agent architecture to the specific task, not through templates or prompt engineering.
>
> (ii) Reuse of optimized collaboration strategies.
> The memory stores optimized $(Q ,C^∗,L,a)$ representing execution-level collaboration strategies rather than textual answers. When similar tasks arise, ALMC retrieves prior collaboration strategies and adapts their structure to the new context. This reflects execution-level reuse based on accumulated procedural knowledge.
>
> In summary,  ALMC achieves adaptivity through the learned synthesis, optimization, and reuse of execution structures, not through static prompts. The dynamic behavior of ALMC arises from execution-level orchestration driven by the LLM’s internal reasoning capabilities, not from prompt engineering.

---

> ### Author Response · Authors · 2025-11-28
>
> > [W3] The claim of “continuous self-improvement” is unsupported. The “RAG memory” is simply text retrieval without validation or analysis of retrieval quality.
>
> We respectfully clarify that ALMC’s memory is not a text-level RAG module and does not retrieve surface strings. Instead, the retriever is integrated into the Solution Optimizer and operates over execution-level collaboration strategies, not textual responses. For each new task, the retriever selects the top-k most relevant past configurations $(Q ,C^∗,L,a)$ based on similarity to the new task instruction, historical execution cost, and Judge-assigned quality scores. These retrieved strategies serve as procedural priors, which the Solution Optimizer and Manager jointly evaluate, adapt, and refine, not as prompt replacements, but as candidate execution plans. This mechanism enables ALMC to improve its execution structure over time: previously successful role-phase-workflow blueprints influence how new tasks are solved, and low-quality or high-cost strategies are naturally discouraged. The improvement is therefore structural, not lexical content.
>
> In `Section 4.3.2` and revised Table R1 (It has been updated in the manuscript), we provide a longitudinal study where ALMC processes each domain dataset in five sequential segments. After each segment, ALMC accumulates new execution strategies in memory and then applies them to the next unseen segment.
>
> Across all domains, performance improves consistently from Seg.1 (cold start) to Seg.5, yielding End-to-Base gains of +4.81 (Code), +2.95 (Medical), +15.00 (Math), and +4.14 (Finance) over the no-memory baseline.
> These results provide direct evidence of continuous self-improvement:
>
> ALMC becomes more effective as it accumulates procedural knowledge, confirming that the JO module supports execution-level learning rather than simple retrieval-based behavior.
>
> **Table R1. Longitudinal Evaluation of JO Learning Process**
>
> | Domain | Baseline (w/o JO) | Seg. 1 (Cold Start) | Seg. 2 | Seg. 3 | Seg. 4 | Seg. 5 (End) | Avg w/ JO | Gain (End-to-Base) |
> |:-------|:----------------:|:-------------------:|:------:|:------:|:------:|:------------:|:---------:|:---------------:|
> | Code | 92.07 | 90.91 | 93.94 | 96.97 | 96.97 | 96.88 | 95.12 | +4.81 |
> | Medical | 78.47 | 77.65 | 79.61 | 80.39 | 81.18 | 81.42 | 80.05 | +2.95 |
> | Mathematical | 70.00 | 65.00 | 70.00 | 75.00 | 75.00 | 85.00 | 74.00 | +15.00 |
> | Finance | 64.04 | 60.87 | 65.22 | 65.22 | 69.57 | 68.18 | 65.79 | +4.14 |

---

> ### Author Response · Authors · 2025-11-28
>
> > [W4] For experiments: The paper repeatedly asserts domain generalization, but the experimental setup only involves standard benchmarks (HumanEval, MedQA, MMLU subsets). There is no transfer or few-shot evaluation proving actual adaptation. The authors compare only against very early/old frameworks.
>
> We would like to clarify that our evaluation already includes true cross-domain transfer: ALMC is tested across coding, medicine, mathematics, finance, and an out-of-domain chemistry split, all of which require adapting workflow structures and collaboration patterns to substantially different reasoning distributions. To further strengthen the evidence, we will additionally include law and astronomy tasks in the revised version, partially presented in Table R2 and `Section 4.2: Out-of-Domain Stress Test`.
>
> **Table R2. Performance comparison across out-of-domain stress tests with ALMC using GPT-3.5-turbo**
>
> | Domain  | Accuracy | Cost/Q [10-4$] | Time/Q [s]  |
> |:---------------------|:--------:|:------------------------------------:|:--------------------------------:|
> | Astronomy | 0.84 | 9.53 | 19.74 |
> | Law | 0.53 | 22.35 | 22.35 |
>
> Regarding the choice of zero-shot evaluation: ALMC is designed to infer roles, phases, and workflow structures without handcrafted domain priors. Few-shot prompting would shift the adaptation burden from workflow synthesis to demonstration imitation, which contradicts the objective of testing emergent collaboration.
> As for baselines, we compare against well-established multi-agent frameworks (Debate, Voting, AgentVerse, ChatDev, DyLAN, etc.)  that form the foundation of most recent systems.
> We agree that incorporating several newer general-purpose frameworks would further strengthen our empirical comparisons, and we will include additional results for CaptainAgent, SPP, and AutoAgent in the revised manuscript (`Appendix C`). We have completed all the comparative experiments (as shown in Table R3).
>
> **Table R3: Performance comparison across four domains**
>
> | Domain | Method | Accuracy / Pass@1 (3.5-turbo / 4o-mini) | Cost/Q [10-4$] (3.5-turbo / 4o-mini) | Time/Q [s] (3.5-turbo / 4o-mini) |
> |:-------|:-------|:---------------------------------------:|:------------------------------------:|:--------------------------------:|
> | **Code** | SPP | 0.55 / 0.74 | 15.24 / 5.49 | 4.70 / 27.07 |
> |  | AutoAgent | 0.25 / 0.69 | 54.88 / 56.71 | 101.63 / 300.67 |
> |  | CaptainAgent | 0.64 / 0.75 | 222.56 / 46.34 | 9.55 / 30.93 |
> |  | **ALMC** | **0.83 / 0.92** | **27.81 / 8.87** | **7.85 / 10.34** |
> | **Medical** | SPP | 0.58 / 0.73 | 13.28 / 4.01 | 4.96 / 11.95 |
> |  | AutoAgent | 0.19 / 0.66 | 85.81 / 47.97 | 181.20 / 196.89 |
> |  | CaptainAgent | 0.61 / 0.74 | 82.12 / 32.33 | 5.86 / 19.34 |
> |  | **ALMC** | **0.64 / 0.79** | **35.37 / 20.15** | **13.09 / 45.88** |
> | **Mathematical** | SPP | 0.35 / 0.64 | 13.00 / 4.00 | 3.76 / 11.10 |
> |  | AutoAgent | 0.20 / 0.45 | 59.00 / 38.00 | 118.18 / 151.87 |
> |  | CaptainAgent | 0.41 / 0.68 | 53.00 / 16.00 | 5.53 / 14.72 |
> |  | **ALMC** | **0.47 / 0.70** | **31.00 / 20.00** | **8.31 / 30.77** |
> | **Finance** | SPP | 0.46 / 0.57 | 14.04 / 3.51 | 4.24 / 9.53 |
> |  | AutoAgent | 0.11 / 0.51 | 47.37 / 42.98 | 73.03 / 191.44 |
> |  | CaptainAgent | 0.44 / 0.64 | 51.75 / 19.30 | 5.12 / 14.30 |
> |  | **ALMC** | **0.53 / 0.64** | **27.19 / 15.79** | **9.64 / 31.61** |

---

### Author Response · Authors · 2025-11-28
**Clarifying ALMC’s Novel Orchestration Capabilities Beyond Existing Multi-Agent Frameworks**

We appreciate the reviewer’s thoughtful comments regarding novelty and gladly clarify the conceptual distinction between ALMC and existing multi-agent approaches.

While a broad range of prior systems explore multi-agent collaboration, they fall into several well-established families, such as role-only team formation (e.g., AgentVerse, CaptainAgent, SPP, AutoAgents), static pipelines (e.g., ChatDev, MedAgents, MDAgents, Magnetic-One), graph-structured controllers (e.g., GPT-Swarm), and argumentation or debate-based frameworks (e.g., DyLAN, Multi-Agent Debate, Vote). These families differ in dynamic behavior, but they share an important structural limitation: none of them provide a dynamic, unified orchestration mechanism, including adaptive team formation, explicit multi-phase decomposition, artifact handling, adaptive execution topology, and experience reuse. Table X summarizes the comparison along five orchestration dimensions.

**Table X: Comparison of representative multi-agent frameworks**
| Framework                   | Adaptive Team Formation | Explicit Multi-Phase Decomposition | Artefact Handoff | Adaptive Execution Topology | Experience Reuse                |
|-----------------------------|--------------------------|-------------------------------------|-------------------|------------------------------|----------------------------------|
| AgentVerse                  | ✔                        | ✘                                   | ✘                 | ✘                            | ✘                                |
| AutoAgents                  | ✔                        | ✘                                   | ✘                 | ✘                            | ✘                                |
| CaptainAgent                | ✔                        | ✘                                   | ✔                 | ✘                            | △ only team caching              |
| SPP                         | ✔                        | ✘                                   | ✘                 | ✘                            | ✘                                |
| ChatDev / Magentic-One      | ✘                        | ✔                                   | ✔                 | ✘                            | ✘                                |
| MedAgents / MDAgents        | ✘                        | ✔                                   | ✘                 | ✘                            | ✘                                |
| GPT-Swarm                   | ✘                        | ✘                                   | ✘                 | ✔                            | ✘                                |
| DyLAN                       | △ select from pool       | ✘                                   | ✘                 | ✔                            | ✘                                |
| Debate / Vote               | ✘                        | ✘                                   | ✘                 | ✘                            | ✘                                |
| **ALMC (Ours)**             | **✔**                    | **✔**                               | **✔**             | **✔**                        | **✔**                            |
**Legend:**
- **✔** = clearly supports this capability
- **✘** = essentially does not support it
- **△** = partial / limited support

In role-only systems dynamically create or recruit experts, but collaboration unfolds as largely free-form group discussion. They lack constraints on when agents should operate, what each stage must produce, or how intermediate artefacts should be propagated. Prior studies [1-6] have shown that such open-ended multi-agent dialogues often suffer from uncontrolled turn-taking, topic drift, overwritten information, and difficulty enforcing responsibilities or task order, making them inherently unstable for multi-stage or dependency-sensitive tasks.

Static pipelines introduce ordered phases, but these phases are entirely hand-crafted and tied to specific domains (e.g., software development or clinical workflows). They cannot adapt to new task types or domains.

Graph-structured controllers optimize routing or reasoning trajectories, but they do not generate or revise agent teams, do not define human-interpretable multi-phase workflows, and do not maintain phase boundaries or preserve artefacts across steps.
Debate or argumentation systems structure opinion exchange, but they do not create or adapt agent teams, it relies on a fixed, manually crafted pool of experts whose prompts and capabilities must be predefined, and the selection process is manual. Thus these systems lack task-driven role generation and cannot support multi-stage or domain-adaptive workflows.

---

> ### Author Response · Authors · 2025-11-28
>
> ALMC introduces a fundamentally different orchestration paradigm.
>
> Rather than coordinating only agents, ALMC coordinates the entire problem-solving process via a dynamically constructed Role-Phase-Workflow representation. This framework governs:
>
>  - who participates (RoleConfig),
>
> - when they participate (PhaseConfig),
>
>  - how information flows and dependencies are organized (WorkflowConfig), and
>
>  - how structured artefacts are preserved, evaluated, and reused across tasks.
>
> This design enables capabilities that no prior category supports:
>
>  - Dynamic phase synthesis rather than fixed or pre-scripted pipelines.
>
>  - Artefact freezing at every phase, ensuring traceability, preventing information drift, and enabling cross-phase consistency checks.
>
>  - Execution-level memory, which stores and reuses entire collaboration structures-not just answers-allowing ALMC to benefit from accumulated experience in ways unavailable to role-only, static, or routing-based systems.
>
>  - A Judge-Optimizer loop that systematically refines roles, phases, and workflow structure based on structured artefacts and execution logs, enabling principled, structure-aware refinement instead of unconstrained conversational heuristics.
>
> These elements collectively allow ALMC to operate not as a group-chat system, nor as a scripted pipeline, nor as a routing engine, but as a dynamic workflow engine capable of long-horizon, multi-stage, and dependency-aware reasoning. We will strengthen the manuscript to make these distinctions clearer, emphasizing that ALMC contributes a new class of multi-agent collaboration that is absent from existing frameworks.
>
> **References：**
>
> [1] Tang, Xiangru, et al. "Medagents: Large language models as collaborators for zero-shot medical reasoning." Findings of the Association for Computational Linguistics: ACL 2024. 2024.
>
> [2] Qian, Chen, et al. "Chatdev: Communicative agents for software development." Proceedings of the 62nd Annual Meeting of the Association for Computational Linguistics (Volume 1: Long Papers). 2024.
>
> [3] Zhuge, Mingchen, et al. "Gptswarm: Language agents as optimizable graphs." Forty-first International Conference on Machine Learning. 2024.
>
> [4] Kim, Yubin, et al. "Mdagents: An adaptive collaboration of llms for medical decision-making." Advances in Neural Information Processing Systems 37 (2024): 79410-79452.
>
> [5] Chen, Weize, et al. "Agentverse: Facilitating multi-agent collaboration and exploring emergent behaviors." The Twelfth International Conference on Learning Representations. 2023.
>
> [6] Yao, Shunyu, et al. "Tree of thoughts: Deliberate problem solving with large language models." Advances in neural information processing systems 36 (2023): 11809-11822.

---

### Author Response · Authors · 2025-12-03
**Summary of Contributions, Strengths, and Revisions (for AC)**

We thank all reviewers for their constructive and insightful feedback, which has significantly improved the clarity and rigor of the manuscript.

**Core contributions.**

ALMC introduces a unified orchestration framework for LLM-based multi-agent collaboration that jointly supports: (1) adaptive team formation with task-specific roles, (2) explicit multi-phase decomposition with structured artifact handoffs, (3) adaptive execution workflow topology, and (4) execution-level experience reuse through a RAG-backed memory. Unlike prior systems that coordinate only at the role or routing level, ALMC orchestrates the entire problem-solving process through a dynamically synthesized Role-Phase-Workflow representation.

**Reviewer-noted strengths.**

(1) Robust empirical performance and cost/latency trade-off analysis, with consistent gains over both general-purpose and domain-specific multi-agent baselines across diverse domains and LLM backbones.

(2) Clear and interpretable architectural design, with explicit roles, phases, and workflow structures supported by diagrams and pseudocode.

(3) Adaptive configuration and reusable procedural experience, addressing the rigidity of predefined roles and static pipelines in existing systems.


**Revisions addressing reviewer concerns.**
- **Novelty clarification**:
We added `Table X (global comment)`, comparing ALMC with AgentVerse, AutoAgents, CaptainAgent, GPT-Swarm, DyLAN, etc. across five core orchestration functions. This clarifies that existing systems coordinate agents, whereas ALMC orchestrates the entire problem-solving pipeline.

- **Evidence of continuous self-improvement**:
We added a longitudinal study over five sequential data segments in each domain, showing consistent performance increases as memory accumulates (`Table R1`). This directly demonstrates experience-driven improvement rather than text-level retrieval.

- **Retrieval transparency and safety**:
Specified indexing fields, similarity threshold ($\tau$=0.7), Top-k ranking ($k$=3), fallback rules, and privacy protections. Added ablations on No-Memory / In-Domain / Cross-Domain retrieval (`Table R7`), verifying stable behavior and the necessity of domain gating.

- **Long-horizon capability**:
We added two complex-task evaluations (`Tables R5-R6`): the ClassEval benchmark (class-level code generation) and full software-application construction tasks (games, tools, and web apps), demonstrating ALMC’s ability to handle iterative refinement and dependency-aware workflows.

- **Stability of pairwise phases and broader comparisons**:
We provided justification grounded in recent literature and included controlled comparisons against Debate, ToT, and GoT under matched budgets (`Table R4`), as well as new evaluations for SPP, AutoAgents, and CaptainAgent (`Table R3`).

---

### Meta-Review · Area_Chair_s5kL · 2025-12-12

**Summary:**

The following concerns remain outstanding and, in my view, prevent the work from being sufficiently solid:
1. The methodological novelty of the proposed approach remains limited.
2. Experimental evaluation on genuinely long-horizon tasks is limited.
2. The empirical evidence supporting claims of "continuous self-improvement" is insufficient.
3. Evidence for domain generalization is not convincing.
5. Comparisons with closely related prior work are not sufficiently thorough.

**Reviewer Concerns:**

**The following concerns have been addressed:**
1. The "adaptive" is learned/optimized or just from prompt engineering.
2. Human cost.
3. Control of backbone, temperature, etc.
4. Specification of fallback behavior when the memory is empty or sparse.
5. Memory structure.


**The following concerns remain outstanding:**
1. The methodological novelty of the proposed approach remains limited.
2. The empirical evidence supporting claims of "continuous self-improvement" is insufficient.
3. Evidence for domain generalization is not convincing.
4. The superiority of pairwise dialog over alternative dialog structures is not adequately justified.
5. Comparisons with closely related prior work are not sufficiently thorough.
6. Experimental evaluation on genuinely long-horizon tasks is limited.

**The following concern remains but is considered minor:**
1. Lack of solid formal theoretial analysis.

**Reviewer Scores:**

1. Reviewer DuXb is expected to mantain the score as 2.
2. Reviewer siHQ is expected to mantain the score as 4.
3. Reviewer o48m is expected to mantain the score as 6.
4. Reviewer K7Lw is expected to mantain the score as 2.

---

### Decision · Program_Chairs · 2026-01-26

Reject